# Last Fifteen Years of Nanotechnology Application with Our Contribute

**DOI:** 10.3390/nano15040265

**Published:** 2025-02-10

**Authors:** Silvana Alfei, Guendalina Zuccari

**Affiliations:** 1Department of Pharmacy (DIFAR), University of Genoa, Via Cembrano 4, 16148 Genoa, Italy; guendalina.zuccari@unige.it; 2Laboratory of Experimental Therapies in Oncology, IRCCS Istituto Giannina Gaslini, Via G. Gaslini 5, 16147 Genoa, Italy

**Keywords:** nanotechnology, nanoparticles (NPs) applications, nano packaging in food, dendrimers, nanosized polymers, liposomes, micelles, nanotoxicology, nanoscience revolution

## Abstract

Currently, nanotechnology is the most promising science, engineering, and technology conducted at the nanoscale (nm), which is used in several sectors. Collectively, nanotechnology is causing a new industrial revolution, and nano-based products are becoming increasingly important for the global market and economy. The interest in nanomaterials has been strongly augmented during the last two decades, and this fact can be easily evaluated by considering the number of studies present in the literature. In November 2024, they accounted for 764,279 experimental studies developed in the years 2009–2024. During such a period, our group contributed to the field of applicative nanotechnology with several experimental and review articles, which we hope could have relevantly enhanced the knowledge of the scientific community. In this new publication, an exhaustive overview regarding the main types of developed nanomaterials, the characterization techniques, and their applications has been discussed. Particular attention has been paid to nanomaterials employed for the enhancement of bioavailability and delivery of bioactive molecules and to those used for ameliorating traditional food packaging. Then, we briefly reviewed our experimental studies on the development of nanoparticles (NPs), dendrimers, micelles, and liposomes for biomedical applications by collecting inherent details in a reader-friendly table. A brief excursus about our reviews on the topic has also been provided, followed by the stinging question of nanotoxicology. Indeed, although the application of nanotechnology translates into a great improvement in the properties of non-nanosized pristine materials, there may still be a not totally predictable risk for humans, animals, and the environment associated with an extensive application of NPs. Nanotoxicology is a science in rapid expansion, but several sneaky risks are not yet fully disclosed. So, the final part of this study discusses the pending issue related to the possible toxic effects of NPs and their impact on customers’ acceptance in a scenario of limited knowledge.

## 1. Introduction

Currently, nanotechnology is the most promising science, engineering, and technology conducted at the nanoscale. Nanotechnology is used in several sectors, and it is included in various merchandise we use every day. The expansion of research, development, and commercialization in nanotechnology has been and is particularly rapid because of the perspective of its significant social, environmental, and economic benefits [1]. Collectively, nanotechnology is causing a new industrial revolution, and nano-based products are becoming increasingly important for the global market and economy [2]. The first important step of the nanotechnology development and acceptance, recognizing its nonpareil potential, was the approval in the US of the National Nanotechnology Initiative (NNI) to promote nanotechnology. Mihail C. Roco proposed the initiative in the year 1999, during a presentation to the White House under the Clinton administration, but it was officially launched in 2000, while it received funding for the first time in Fiscal Year (FY) 2001 [3]. Specifically, in July 2000, Neal Lane, Assistant to the President for Science and Technology, signed the NNI Implementation Plan for the program with the first annual budget of $464 million [3]. Collectively, the NNI budget increased from about $270 million in 2000 to about $1.9 billion in 2010, with an average growth rate of about 21.5%. The NNI investment has been relatively stable between 2010 and 2020, changing in the quality of the contents. NNI cumulative investment by 2023 reached $40 billion, and nanotechnology has become pervasive in material, energy, and biosystem-related discoveries and applications [3]. Then, considerable attention has been paid to nanotechnology development also in Japan and western European countries, such as Germany, the United States, and France. More recently, countries in the Asia-Pacific region have also begun to make significant progress in nanotechnology research and development [2]. National Research and Development (R&D) programs on nanotechnology were in fact announced by Japan (April 2001), South Korea (July 2001), the European Community (March 2002), Germany (May 2002), China (2002), and Taiwan (September 2002). International dimensions of nanotechnology application have become evident when about 80 countries worked out nanotechnology activities by 2005, partially inspired by the NNI [3]. The growing demand for nano-engineered and nano-enhanced products in various sectors has significantly influenced the productive programs of several industries, and millions of new jobs were created [4]. In 2001, the global profit from products incorporating nanomaterials in seven sectors of the economy was estimated to reach $1 trillion by 2015 [5]. The estimated global revenue of $1 trillion was instead reached already in the year 2013, of which about $284 billion were in the United States. Collectively, the United States had about 33% ($110 billion) of the world incomes of $335 billion in 2010, 33% in 2011, 28% in 2013, and 25% ($750 billion) of the world revenues of $3 trillion in 2020. Particularly, the US nanotechnology-related profit would be about 3.6% of national Gross Domestic Product (GDP) ($20.94 trillion) in 2020 [3]. Nanotechnology was predicted to have influenced the global economy by more than $3 trillion as of 2021, with nanotechnology-based industries worldwide requiring more than 6 million workers to support them [6]. Well-known companies have incorporated nano-based products to improve their market competitiveness [7]. Despite the observed rapid increase in the number of patents related to nanotechnology in the last two decades (STATNANO: Nano Science, Technology, and Industry Information, available online at https://statnano.com/), and in the patent-to-nominal GDP ratio around the world from 2001 to 2019, these data are only an indirect measure of the economic contribution of nanotechnology [7]. Nevertheless, measuring the direct contribution of nanotechnology to the economy is challenging since it is hardly quantifiable and can be measured only vaguely [1].

### 1.1. Application of Nanotechnology Across Industries

Currently, nanotechnology is extensively worked on in several industries, and numerous nanoproducts are being produced. Moreover, in the countries using nanotechnology, it has legislative support. The number of companies using nanotechnology to produce their commodity can be taken as a measure of the nanotechnology’s commercialization benefits. According to data from the Statnano website (STATNANO: Nano Science, Technology, and Industry Information, available online at https://statnano.com/) of June 2021, 3071 companies from 455 countries use nanotechnology to produce various types of products, with a total number of 9157 nanotechnologic commodities (Figure 1).

Electronics have the largest number of products (1931) among the various industries (106) from different countries (18), followed by medicine (1114, 406, 44), including the sector of mRNA vaccines [8]. The most significant economic contributors are the textile and construction sectors, which employ nanotechnology in 42 and 41 different nations, respectively. The agricultural sector also plays a crucial role in developing and using nano products. Anyway, the number of nano products, companies, and countries using nanotechnology is increasing over time [1]. In terms of economic impact and profits, semiconductors with a size under 100 nm represented about 60% (or $90 billion) of the total revenues of $150 billion from all semiconductors in 2010 and about 80–90% (or $350 billion) of total profits of $410 billion from all semiconductors in 2020 [3,8]. The penetration rate of nanotechnology in key industries seems to correlate with the percentage of overall spending on R&D in the respective industry [3].

### 1.2. Fifteen Years of Nanoparticles in the Literature

The interest in nanomaterials has strongly increased during the last two decades and can be easily evaluated by considering the number of studies present in the literature. From a survey conducted using the Scopus database (https://www.scopus.com/, accessed on 6 November 2024), the number of experimental studies intended as full articles, conference papers, and letters, found using the keyword “nanoparticles”, published in the last fifteen years (2009–2024), was 764,279 (Figure 2, light blue line).

That found summing the results obtained by using as keywords “antimicrobial AND nanoparticles” and then “antibacterial AND nanoparticles” was 82,286 (Figure 2 and Figure 3, blue lines).

That found summing the results obtained by using “anticancer AND nanoparticles”, “antitumor AND nanoparticles”, and then “antitumour AND nanoparticles” resulted in 42,390 (Figure 2 and Figure 3, red lines), that obtained using “biomedical AND nanoparticles” was 24,056 (Figure 2 and Figure 3, purple lines), that using “environment AND nanoparticles” was 42,845 (Figure 2 and Figure 3, green lines), and finally that obtained using “nanomedicine” was 21,555 (Figure 2 and Figure 3, pink lines). As observable in Figure 2, the number of studies published on NPs (764,279) is very large, and the sum of publications dealing more specifically with definite applications of NPs, considered in Figure 2 and Figure 3 (213,132), does not reach it, thus leaving a great clean space between the light blue line of NPs and all other lines grouped below. This establishes how the number of NPs applications missing in the graph is high. Particularly, among the applications of NPs randomly chosen and considered in the graphs of Figure 2 and Figure 3, those dealing with human health and medicine (170,287) are almost 4-fold more numerous than those dealing with the environment (42,845). However, several sectors of both these areas have not been reported, including those of food, food packaging, diagnosis, imaging, sensors, electric and electronic components, thermal and electric conductors, fertilizers, pesticides, soil improvers, and so on. They would account for 225,129 other publications, for a total of only 57% of publications found on nanomaterials. During the period reported in Figure 2 and Figure 3, our group contributed to the field of applicative nanotechnology with several experimental and review articles, which we hope could have relevantly enhanced the knowledge of the scientific community in the field. In this new study, the main types of NPs, the methods employed to prepare them, the most adopted technique for their characterization, and their applications in the medical and food sectors developed so far have been extensively discussed. Particular attention has been paid to nanomaterials developed to enhance the bioavailability and target delivery of bioactive molecules, to reduce their possible toxicity, and to nanomaterials used for ameliorating the traditional food packaging. Then, we have reviewed our experimental study on nanosized materials, both in the form of nanoparticles (NPs), dendrimers, micelles, and liposomes developed in the years 2009–2024 [9,10,11,12,13,14,15,16,17,18,19,20,21,22,23,24,25,26,27,28,29,30,31,32,33,34,35,36,37,38,39,40,41,42,43,44,45,46,47,48,49,50,51,52,53,54,55,56,57,58,59]. Additionally, a list of review articles composed by us along the same years and concerning the same topics is particularly useful because collecting the most relevant advances made by other eminent scientists in nanotechnology has been reported [60,61,62,63,64,65,66,67,68,69]. Anyway, paradoxically, although the application of nanotechnology translates into a great improvement in the properties of pristine materials, there may be a still not predictable risk for humans, animals, and the environment associated with an extensive application of NPs. Unfortunately, despite nanotoxicology being a science in large expansion, such risks are not yet fully disclosed. In this state, the final part of this study discusses the pending issue relating to the possible toxic effect of NPs and their impact on customers’ acceptance in a scenario of limited knowledge.

### 1.3. Methods

To select the literature material useful to edit this review, we used both keywords like those reported in the keywords section and those reported in the captions of Figure 2 and Figure 3. Specifically, we inserted such keywords in the Scopus, PubMed, and PubChem databases, finding several reviews and experimental studies, whose number was immediately reduced by removing duplicates. The remaining studies were further reduced to the 248 references used in this study by removing the obsolete ones and then by dividing the residuals into experimental and review studies. The review study was used to obtain the most updated information and recent advances concerning the main types of nanomaterials developed in the last 15 years, the methods to produce them, the techniques for their characterization, and their performances. The review articles were also used to get information on the advantages and disadvantages of the different types of NPs in terms of biomedical applications and mechanical properties. Differently, the experimental studies were used to take information about the applications of nanomaterials, as enhancers for bioactive synthetic and natural compounds, including food constituents, phytochemicals, and nutraceuticals, and as reinforcers in food packaging. Additionally, the experimental study provided us with useful data to organize the tables, including that containing information about our experimental achievements in the applications of nanoparticles, micelles, liposomes, and dendrimers.

## 2. Among the Nanotechnology Applications

### 2.1. Application of Nanomaterials to Natural and Synthetic Bioactive Molecules

Before passing on reviewing our fifteen years of working in the field of nanotechnology, an exhaustive overview about its employment in the medicine and food sectors has been provided in the following sections, with particular attention to the use of nanotechnology to improve the characteristics, often not favorable, of bioactive principles (APs).

Anyway, as shown in Figure 1, it is necessary to keep in mind that the sectors in which nanotechnology is currently applied are several and are not limited to those reported in Figure 2 and Figure 3, and even less so to those detailed in the following sections and/or studied by our group. As recent examples, alternative applications in the use of nanomaterials concern the development of quantum technologies [70] or the production of smart coating surfaces with enhanced antimicrobial properties [71]. The development of quantum technologies in the field of magnetic force microscopy can significantly make easier the handling and processing of the acquired magnetic data [70]. Magnetism plays a pivotal role in many biological systems. However, the intensity of the magnetic forces exerted between magnetic bodies is usually low. In this regard, the development of ultra-sensitivity tools for proper sensing is urgently needed. In this framework, novel micro- and nanofabrication procedures for magnetic force microscopy (MFM) tips, which could enhance the magnetic response signal of tested biomaterials, have been described [70]. In this study, some relevant examples, where MFM can quantitatively assess the magnetic performance of nanomaterials involved in biological systems, including magnetotactic bacteria, cryptochrome flavoproteins, and magnetic nanoparticles that can interact with animal tissues, have been depicted [70].

Moreover, magnetic NPs are currently excellent candidates to address the need for alternative antimicrobial agents capable of overcoming the worldwide worrying problem of antimicrobial resistance (AMR). Particularly, iron oxide nanoparticles (IONPs) are among the most promising antibacterial agents of a new generation, due to their unique magnetic properties and excellent biocompatibility, despite a few disadvantages [71]. The primary mechanism of action of IONPs is through reactive oxygen species (ROSs) induction. IONP characteristics, including their size, shape, surface charge, and superparamagnetism, strongly influence their biodistribution and antibacterial activity. Additionally, external magnetic fields, foreign metal doping, and surface, size, and shape modification can be tuned to improve their antibacterial effects [71].

Anyway, nanotechnology, the different techniques of nano-formulation, and nanomaterials are strongly implied in the current methods used to address the drawbacks concerning bioactive molecules bioavailability [72]. The solubility, delivery, and cell uptake of APs can be strongly improved by using NPs, as well as their protection from early degradation and fast metabolism. In this context, although the existence of several critical challenges, including reproducibility, proper characterization, and biological evaluation via proper assays, are still associated with their use, the Food and Drug Administration (FDA) and the European Medicines Agency (EMA) have approved several nanomedicines, which are now commercially available [72]. Rigorous studies, besides stringent guidelines, are warranted for effective and safe nanomedicine development and use [72]. Moreover, by the nanotechnological manipulation of bioactive molecules, it is possible to prepare food products enriched with them and therefore with improved health properties without interfering with the sensory and quality properties of the original food. It is foreseen that the market for nanotechnological items produced in the food and beverage sectors as health promoters will be incessantly increasing [73]. The nanomanipulation of APs, regardless of their natural or synthetic origin, can allow them to more easily bypass the physiological barriers that commonly limit their oral delivery. It is forecast that synthetic nanomedicines will have nonpareil advantages in drug delivery, as well as in clinical practice in the future [74]. Several factors could, in fact, affect oral absorption of APs, including poor aqueous solubility and therefore a slow dissolution rate in gastrointestinal (GIT) fluids, instability in the acidic environment of the stomach, the presence of degrading enzymes in GIT, the presence of food, biological barriers, and finally, first-pass metabolism in the liver [75]. Moreover, also when systemic circles and/or cells are reached, other issues could consist of the tendency of APs to bind irreversibly to blood proteins. Furthermore, APs could tie permanently to cellular DNA and proteins or form weakly soluble complexes with calcium and magnesium ions, which greatly reduce transcellular absorption [76], thus reducing their health effects. The lack of pathogenesis-targeting effects in neurodegenerative diseases such as Parkinson disease (PD), Alzheimer disease (AD), the various forms of sclerosis, and dementia is principally due to the limiting effects of the blood–brain barrier (BBB), which keeps out of the brain about 99% of all “foreign substances” [77]. Nanotechnology, when correctly applied to drugs suffering from the abovementioned drawbacks, can enhance their efficacy and in vivo stability while reducing their toxicity, thus aligning the excellent results commonly observed in vitro with those found in vivo, which are usually much less satisfactory [62]. Carrying bioactive compounds in NPs favors their distribution in specific brain areas, thus providing more valuable benefits in neuro-regenerative treatments, while minimizing their accumulation in the systemic circulation, as well as the related toxic side effects [78]. Specifically, by loading neurotrophin in NPs, its distribution in specific brain areas has been favored, thus providing more valuable benefits in different types of neuro-regenerative treatments [78]. Moreover, NPs formulation of APs protects them from early degradation and rapid metabolism. Several are the natural APs extracted from fruits, seeds, and vegetables, which are endowed with several health-promoting properties. These activities of natural APs have been extensively improved by engineering using nanotechnology, thus ameliorating their very poor solubility, as well as the many pharmacokinetic drawbacks associated with their pristine form [62]. Several studies have reported on the development of appropriate nanomaterial-based devices used to enhance the solubility of strong antioxidants, polyphenols, their hydrophilic–lipophilic balance (HLB), GIT absorbability, and/or to protect them from early oxidation and/or metabolism [19,79,80,81,82,83,84]. It is the case of the insoluble ellagic acid (EA), which was gifted with high water solubility using cyclodextrins [82,83,84], pectin [19], and polyester-based dendrimers [19]. In a paper, the effects of phospholipid composition on the pharmacokinetics and biodistribution of epirubicin-loaded liposomes were examined, proving a significantly prolonged circulating time, reduced clearance, and reduced heart toxicity [85].

#### 2.1.1. NPs-Mediated Controlled Release of APs

The controlled and targeted release of APs from NPs has been recognized as a pivotal step for realizing their effective administration. A controlled and targeted delivery of APs allows them to reach their higher concentration at the desired site, thus permitting a reduction in the overall dosage and the systemic toxicity affecting the pristine AP. Many are the parameters that can be optimized to control the specific release of APs, including pH, temperature, ultrasound, or magnetic field applications; light incidence, type, and physicochemical features of NPs; chemical structure; and physicochemical features [86].

APs-loaded stimuli-sensitive nano-capsules possessing an oil *core* were shown to improve the effects following the oral administration of pristine APs, while the dose and the administration frequency, thus ameliorating the patient compliance, were reduced [86].

Rhodamine-loaded poly-alkylene glycol (PAG)-NPs were applied to SH-SY5Y NB cells or prostate cancer DU145 cells and were visualized by fluorescence. PAG-NPs were visualized in the cytoplasm, suggesting that they have been internalized via endocytosis, overcoming, without damage, the phospholipidic barrier of the cell membrane, which represents an impediment for hydrophilic compounds to enter the cells [85].

#### 2.1.2. Main NPs Developed to Nano Formulate Natural and Synthetic APs

To provide readers with several pieces of information about the main NPs developed to nano formulate natural and synthetic APs, using a tool as reader-friendly as possible, the details on this topic have been organized in Table 1, which summarizes the most used engineered NPs for biomedical uses and/or in the food sector. Furthermore, as an example, the following Figure 4 visualizes the nano-formulation routes reported in the literature for lipophilic antitumoral APs, such as paclitaxel (PTX) or doxorubicin (DOX) [87]. By these methods, it is possible to achieve nanosized dosage forms for more efficient anticancer therapy.

The conventional techniques to prepare NSs consist of the bottom-up and top-down methods. Figure A1, in Appendix A, provides a schematic representation of both techniques, and additional information has been included in Figure A1 caption [69].

β-carotene was formulated as nanomaterial by precipitation from pressurized ethyl acetate-on-water emulsions for application as a natural colorant [88]. Quercetin was instead subjected to high-pressure homogenization (HPH), achieving an NS of amorphous NPs [89]. Using the spray drying (SD) technique and maltodextrin as an encapsulating agent, water-re-dispersible powders loaded with the products derived from acai fruit were prepared. They demonstrated improved nutritional values, extended shelf life, and radical scavenging activity [90]. A stable aqueous NPs (150 nm) suspension of α-tocopherol, with improved solubility and bioavailability, was obtained by a supercritical assisted process [91]. To improve the performances of the single approaches, combination methods were born by merging the top-down and bottom-up techniques. They include the Nanoedge™ Technique (Baxter Healthcare, Lurago D’erba, Como, Italy) [60,131], H 69 Technology, H 42 Technology, H 96 Technology [60], and Combination Technology (CT)

CT, which is suitable for scaling up [132], was used to formulate hesperidin. In vitro studies have established its antioxidant activity, and when assumed with the diet, it has proven to be a valid vase protector. NPs were characterized by improved solubility and long-term stability. They were suitable both for oral administration and topical application [60]. Hesperidin nanocrystals are in the Platinum Rare cosmetic product (La Prairie, Volketswil, Switzerland). Furthermore, rutin and apigenin were processed with the CT technology. Rutin NPs of about 600 nm, suitable both for oral and topical administration, and apigenin NPs of 275 nm were obtained. Rutin nanocrystals can be found in a cosmetic product launched by Juvena, St. Margrethen, Switzerland [60].

Nanoedge-like techniques were employed to formulate all-*trans* retinoic acid (ATRA) in 155 nm sized particles, suitable for oral administration, in 30′ operation time [60].

H 69 Technology (SmartCrystal^®^ technology group, GfN Selco, Wald-Michelbach, Germany) was approached to formulate resveratrol (RES) in particles of 150 nm, suitable for oral administration [69].

H 42 Technology is like H 69 [131]. When H 42 was used for formulating RES, particles of 200 nm eligible for oral administration were obtained [60,131]. Differently, through NEs technology, APs are encapsulated in small droplets mixing an aqueous phase (w) with an oil one (o) and obtaining water in oil (w/o), oil in water (o/w), or bi-continuous colloidal dispersions. The colloids are stabilized using specific additives, such as generally regarded as safe (GRAS) pharmaceutical surfactants, co-surfactants, and emulsifiers (5–10%). Oils utilized in NEs encompass Captex 355, Captex 8000, Witepsol, Myritol 318, Isopropyl myristate, Capryol 90, Sefsol-218, triacetin, isopropyl myristate, castor oil, olive oil, etc. In bi-continuous colloidal dispersions, microdomains of oil and water are inter-dispersed in the system.

NEs can be achieved by high- and low-energy methods, such as high-pressure homogenization, ultrasonication, phase inversion temperature, and emulsion inversion point, as well as recently developed approaches such as bubble bursting methods [92]. High drug loading (DL%) is possible, and solutions are isotropic, transparent, and kinetically stable, even if NEs stability is lower than that of microemulsions, due to the very small droplets initially obtained, which tend to re-aggregate along time with the formation and growth of undesired great crystals [133,134,135]. Using NEs-based delivery systems, herbal drugs, whole plant extracts, or their constituents, as well as food-related APs, unstable in highly acidic pH and/or undergoing liver metabolism if administered as free, were formulated. NE techniques were considered to reduce possible side effects due to the accumulation of some APs in the non-targeted areas. For this characteristic, NEs are also authorized for pediatric and geriatric oral administration [133].

Self-emulsifying drug delivery systems (SEDDSs) represent a particular type of NE. SEDDSs are generally suitable for orally delivering LBACs and include SNEDDSs and SMEDDSs based on their droplet size [60]. They can be taken orally by either solubilizing them in water and drinking the obtained NE or by ingesting capsules filled with gelatin, as schematized in Figure A2 in Appendix A.

Many formulation parameters, including surfactant concentrations, the oil/surfactant ratio, the polarity of the emulsion, the droplets’ size and charge, and the physicochemical properties of APs, such as pKa, log P, molecular structure, MW, presence, and quantity of ionizable groups, have remarkable effects on the performances of SEDDSs.

The group of Hu manufactured a self-double-emulsifying drug delivery system (SDEDDS) loaded with epigallocatechin-3-gallate (EGCG), having improved photo-stability in respect of free EGCG [100].

NE techniques were used to nano-formulate turmeric, curcumin (diferuloylmethane), and di-benzoyl-methane (a structural analogue of curcumin). Curcumin, also used as a GRAS food supplement, possesses antiseptic, analgesic, antimalarial, and insect-repellent activities. Triacylglycerol was chosen as the oil phase and Tween-20 as an emulsifier to formulate curcumin in NE, achieving NPs with reduced toxicity, improved bioavailability and bioactivity, and strong anti-inflammatory properties [93]. Turmeric is instead commonly used to treat biliary disorders, jaundice, anorexia, cough, diabetic ulcers, liver disorders, rheumatism, inflammation, sinusitis, menstrual disorders, hematuria, and hemorrhage [60]. Furthermore, tannins, stilbenes, and flavonoids, possessing at least in vitro antioxidant effects, have been encapsulated in Nes [94]. Differently from in vitro results, the in vivo antioxidant activity shown by EGCG was very poor. However, it was significantly increased by formulating it in small NPs using the NE technique [95]. Bioactive lipids and carotenoids were converted in Nes, achieving, respectively, more stability against autoxidation and increased bio-accessibility [60]. NEs were also successful in saving lactic acid bacteria from degradation and in restoring the proper microbiota in diverse intestinal disease conditions [136].

Pomegranate peel ethyl acetate extracts containing several polyphenols, including high levels of ellagic acid (EA), were beat together with pomegranate seed oil, achieving polyphenol-loaded NEs, suitable for topical applications. The NE possessed the capability to avoid or delay UV radiation damage, thus being suitable as an anti-photo-aging cosmetic [60,96]. Lemongrass essential oil (LEO), often found in soaps and other personal care products as flavor, is traditionally used to treat digestive problems and high blood pressure, as a tool in aromatherapy to relieve stress, anxiety, and depression, or like an antimicrobial. Unfortunately, LEO is prone to autooxidation and easy degradation, by which it loses activity and provides smelly or even harmful compounds, responsible for allergic reactions and skin irritation.

To address such drawbacks, NE formulations of LEO were prepared with reduced undesired sensory impact while enhancing its antimicrobial activity. Edible carnauba wax and LEO NEs were developed, achieving a coat packaging for protecting plums, which proved to inhibit the growth of food-borne *Salmonella* spp. and *E. coli* [97].

By using APs-loaded solid nanoparticles delivery systems (SNDSs), highly soluble bioactive nanomaterials were obtained. Further details are reported in Table 1 [101,102]. The SNDSs’ digestibility in the GIT or in other body districts controls the release of APs in that precise body area, thus realizing a targeted release. In this regard, materials of carrier agents should be selected based on their physicochemical features, which should be appropriate to permit SNDSs degradation where desired [101]. Starch-based NPs are digested at an oral level by the activity of amylase, while polysaccharide- and protein/polysaccharide-based NPs are assimilated in the small intestine due to variations in pH and salt concentrations [103]. According to these degradative processes, the APs release happens in the oral cavity from starch-based NPs, while in the small intestine, it happens from polysaccharide and protein/polysaccharide-based NPs. In contrast, lipid-based NPs will release APs in the small intestine simultaneously with the digestion of triglycerides [101].

Particle sizes of SNDSs can positively influence the transport of APs through enterocytes by transcellular endocytosis, while their surface charge could be responsible for the formation of hydrogen bonds with the mucosal surfaces, contributing to momentary retention [137]. On the other hand, the presence of surface cell-penetrating ligands could enhance transmembrane transport efficiency [101], thus positively influencing the effectiveness and bioactivity of the transported APs. Additionally, NPs equipped with a lipid phase can access the bloodstream via mesenteric lymph and thoracic ducts, avoiding hepatic first-pass metabolism, thus extending the half-life of APs-loaded SNDSs.

APs-loaded SNDSs were prepared in the form of micelles (MICs) using some of the polymers reported in Table 1 (row 4). They are present in many therapeutic devices approved by the Food and Drug Administration (FDA) or in clinical trials Phases II–IV [104].

Different APs-loaded SNDSs were also prepared also in the form of hydrogels using different polymers or co-polymers, including PCL-b-PEG-b-PCL (10 nm), PLGA-b-PEG-b-PLGA (77–84 nm), PLA-b-PEG (<200 nm), Pluronics^®^ (<60 nm), PGA-b-PAE (100–200 nm), PLL-b-DOCA-b-mPEG (<200 nm), PEG-b-Pasp (22 to 60 nm), PLH-b-PEG (112 nm), PEI-g-PVP (142 nm), PDMAEMA-PCL (<150 nm), PEG-b-PLL-b-PLLeu (100–125 nm), PIHCA-Tween80 (<320 nm), sodium alginate-HPMC, PEO-b-PHB-b-PEO, OncoGelTM, and PAH/Chitosan, and are already approved and marketed as medical treatments for different diseases [104]. Interestingly, regarding dendrimers characterized by a *core–shell* structure, when APs were physically encapsulated, the resulting AP-enriched dendrimers were characterized by having a bioactive functional *core* and a dendrimer *shell*. In contrast, when APs were covalently bound on the surface of dendrimers, the dendrimer formulations were typified by a dendrimer *core* and a bioactive *shell*. The drug-loaded dendrimers showed a favorable drug release profile protracted in time and improved biological activities.

More specifically, with organic solid nanoparticles (OSNPs), organic nanocarriers are intended, whose classification is based on their physicochemical nature, production method, properties, free energy, interaction type, typology, etc. [138]. To date, the most adopted OSNPs for APs encapsulation are those reported in Table 1. Anyway, inorganic metal oxide-based and clay-based NPs are also extensively used.

The main typologies of lipid-based nanoparticles (LNPs) have been reported both in Table 1 and in Figure A3 in Appendix A [139].

SLNPs are emerging products of lipid nanotechnology [140,141], are commercially available NPs, and are suitable for delivering LAPs. The lipids used to obtain SLNPs include triglycerides (tristearin), diglycerides (glycerol bahenate), monoglycerides (glycerol monostearate), fatty acids (stearic acid), steroid molecules (cholesterol), and waxes (cetyl palmitate) [60].

LPs have demonstrated remarkable therapeutic benefits in clinical applications, even if their approval is still limited by all stages necessary for the liposomal development and for the production process that encompasses manufacturing methods, regulatory approval by the competent authorities, and intellectual property [142]. Anyway, due to intensive research in the development of liposomal formulations for clinical use, a few liposomes have entered the market as commercialized liposomal products [143]. The main liposomal products approved and marketed have been reported in Figure 5 [143], while the main APs contained in such clinically approved liposomal formulations have been reported in Table 1 [106].

Anyway, even if not marketed until now, several other or similar APs have already been formulated in several other liposome carriers and are currently in clinical trial Phase I–III. Those that are in Phase III clinical trials include APs such as amikacin, tecemotide, T4 endonuclease V, prostaglandin E-1 (PGE-1), doxorubicin, and cisplatin. Those in Phase II encompass platinum analogue *cis*-(*trans*-R,R-1,2-diaminocyclohexane) *bis* (neo-decanoato), platinum (II) semi-synthetic doxorubicin analogue, annamycin, cisplatin, lurtotecan, potent topoisomerase I inhibitor, irinotecan’s active metabolite, paclitaxel, and ATRA. Finally, those in Phase I include APs such as mitoxantrone, antisense oligodeoxynucleotide growth factor receptor bound protein 2 (Grb-2), vinorelbine tartrate, topotecan, PLK1 siRNA, PKN3 siRNA, doxorubicin, CEBPA siRNA, docetaxel, cisplatin, doxorubicin, p53 gene, and vinorelbine [106]. Furthermore, LNPs have been used to entrap essential oils (EOs), ferulic acid, and tocopherol, achieving loaded lipid NPs, which showed the capability of reaching different types of cells and improved antioxidant activity [144]. Micelles (MICs) are tiny spherical lipid particles made using both hydrophilic and hydrophobic copolymers like those reported for SDDSs. Usually, PEG-PLGA micelles are normal micelles (n-MICs), while PLC-P2VP micelles are inverse micelles (i-MICs). MICs-based delivery systems allow the intravenous administration of HAPs without using solubilizing adjuvants, which can cause undesired toxic symptoms [107]. The release of APs from MICs can be voluntarily provoked at the target site by local stimuli, such as variation in pH, temperature, or the application of ultrasounds or light. Drug-loaded MICs found applications, especially in the treatment of cancer disease, and the selection of stimuli-sensitive polymers used in MIC preparation is based on the specific conditions found in the tumor microenvironment [107]. The current clinical trends in using stimuli-responsive MICs to treat cancer have been reported and discussed in a relevant study by Wang et al. [107]. Stimuli include pH, ROS, hypoxia, enzymes, thermic, and magnetic stimuli [107].

Anyway, many factors, including MIC intrinsic stability, APs diffusion rate, their partition coefficient, the copolymers biodegradation rate, APs concentration within the MICs, their MW, physicochemical features, and location within the MICs, can also influence their release [60].

Fatty acid-based micelles were used to solubilize and transport plant oxylipins, phytoprostanes, and phytofurans, which were derived by the non-enzymatic oxidation of linolenic acid [101].

Niosomes (NIOs) are osmotically active and stable vesicles, representing an alternative option to LPs specifically used for ameliorating oral bioavailability of APs with limited absorption in GIT. They can act as reservoir systems capable of providing controlled and sustained delivery of encapsulated APs. NIOs made with non-ionic surfactants demonstrated low levels of toxicity for cells due to their uncharged structure [108]. NIOs, as MICs and LPs, have been employed to formulate APs such as those reported in Table 1, clinically applied to treat different forms of cancer, including breast, lung, colorectal, prostate, and skin cancer [108]. By mixing Span 60 and Tween 60 with 15% PEG 400 as a solvent, a dermal delivery system consisting of EA-loaded NIOs was prepared. It exhibited very high EE% and high efficacy in delivering EA to human epidermis and dermis [145].

ONPs consist mainly of cyclodextrins (CDs), which are commonly used as host molecules for encapsulating and delivering LAPs by the monomolecular inclusion complex technique [114,115]. CDs are cyclic oligosaccharides of different dimensions obtained through enzymatic degradation of amylose by the enzyme cyclodextrin glucosyl transferase [60]. They have a truncated cone structure and can accommodate hydrophobic molecules inside their hydrophobic interior cavity. CDs’ outer side, due to the presence of several OH groups, forms a hydrophilic layer, which confers CDs high water solubility (Figure A4 in Appendix A).

Low doses of CDs are well tolerated by humans, but high doses may cause some adverse effects such as diarrhea and soft stools. β-CDs are currently mostly used as devices for drug delivery and loading several non-polar APs [113] and found applications as carriers in the food, pharmaceutical, and cosmetic industries. Different methods are available to prepare the inclusion complexes (ICPXs) of LAPs using CDs [114,115].

Most studies assert that by encapsulation in CDs, significant improvements were observed in polyphenols, such as flavonoids or other APs from plants, including those reported in Table 1 [60,69,116,117,118]. The improvements concerned mainly ameliorated water solubility, water dispersibility, stability, antioxidant and anti-inflammatory activity, drug loading (DL%), controlled release, and oral bioavailability, while possible bitter taste perception and degradation were reduced.

Polysaccharide NPs (PNPs) are instead synthesized from natural polyelectrolytes or non-electrolytes, hydrophilic polysaccharides such as alginate, chitosan, hyaluronic acid, pectin, and cellulose derivatives (hydroxyethyl cellulose and carboxymethylcellulose), and proper cross-linkers or other substances inducing polymer–polymer interactions, as schematized in Figure A5 in Appendix A.

APs were either physically entrapped during NPs formation, covalently attached to the precursor materials, or absorbed into NPs after their preparation. PNPs can be freeze-dried (FD) in the presence of a suitable cryoprotectant or spray-dried (SD) into a microparticulate powder. PNPs have a high affinity for mucosal layers of the cells present in the respiratory tract and GIT, thus being capable of long residence time in these districts. Moreover, their biodegradability, biocompatibility, mucoadhesive features, and tunable properties make them attractive as carriers for formulating colon-targeted drug delivery systems [146]. PNPs, mainly those made using cationic chitosan, anionic alginate, or combinations of alginate and chitosan, allowed the administration of several APs, also food-derived, for treating diseases in several body compartments such as nasal, oral, ocular, and dermal with enhanced circulation time [69]. Anionic hyaluronic acid-based NPs are particularly efficient for targeting delivery of anticancer drugs, due to hyaluronic acid’s affinity for hyaluronan receptors, which are highly expressed in tumor cells. Finally, neutral PNPs, made of dextran, maltodextrin, pullulan, and pectin, were used to prepare delivery systems able to escape the reticuloendothelial system, thus possessing long systemic residence time, circulation permanence, and higher efficiency [69]. Maltodextrin is digested like glucose and is massively used by the bodybuilding industry to increase the intake of carbohydrates in the diet without resorting to sugar [69]. Some plant extracts and APs encapsulated in PNPs are listed in Table 1 [119,120,121,122,123,124,125,126,127,128,129,130].

Protein-based NPs (ProNPs) can be prepared through protein precipitation methods, including de-solvation, coacervation, emulsification, nanoprecipitation, SD, NP albumin-bound technology, self-assembly, electro-spraying, salting out, and cross-linking [69]. Specifically, proteins are dissolved in a suitable solvent, and a non-solvent is added. Also, by changing the physicochemical parameters of the protein solution (pH, salinity, or temperature), precipitation of the pristine protein can be caused [109]. De-solvating agents are often added to promote the dehydration of the system. The stability of the pristine protein is increased using chemical, ionic, thermal, and enzymatic cross-linking agents, among which 8% glutaraldehyde aqueous solution or calcium phosphate are the most used. Figure A6 in Appendix A shows a casein protein complex stabilized with calcium phosphate [69].

An innovative method was introduced, which allows to produce cross-linked and sterilized ProNPs in a one-step procedure based on γ-irradiation of ProNPs in phosphate buffer (pH = 7.2) in the absence and/or presence of ethanol and methanol at 30% and 40% (*v*/*v*). The results showed that by controlling the irradiation dose, it was possible to modulate the cross-linking density and the particle size [110]. Moreover, to enhance their circulation residence time, ProNPs have been surface-modified with PEG [60].

Several food-related APs have been formulated using ProNPs, some of which have been included in Table 1, such as EGCG, GA, and probiotic microorganisms [60,111].

Concerning organo-synthetic biodegradable polymer nanoparticles (OBP-NPs), they are already described when SNPs made with biodegradable polymers were discussed. They can load different APs, either by physical interactions or by covalently binding by utilizing their several chemical functions. Depending on the hydrophilic/hydrophobic balance (HLB) of polymers or copolymers, NPs characterized by various shapes and morphologies can be prepared.

They are also suitable for oral administration of nutraceuticals and phytochemicals and for producing food-grade smart nanocomposites for food packaging (FP), able to preserve food quality, looks, and taste along with storage.

In the food industry, a topical ointment was prepared with PEG and 5% pomegranate rind extract, with an excellent release profile and skin-permeation capability of EA and anti-inflammatory effects in a mouse model of contact dermatitis [147,148]. NPs (150–300 nm) made of PLGA, chitosan, and PEG were loaded with EA (up to 100 μM), achieving EA-loaded PLGA-chitosan-PEG NPs that were able to potentiate apoptosis-mediated cell death in HepG2 human hepatoma cells [149]. PLGA-based NPs stabilized by PEG were used to encapsulate anthocyanins, obtaining anthocyanins-loaded biodegradable NPs that showed an EE% of 60%, improved stability, extended life, and a biphasic release profile in vitro. In vivo, they proved anti-inflammatory and anti-neurodegenerative capacities, preventing memory losses in estrogen-deficient rats, and showed a neuroprotective power against Alzheimer’s dementia [150,151,152]. Finally, anthocyanins formulated as NPs significantly upregulated endogenous antioxidant genes, thus helping in the prevention of oxidative stress (OS), with consequent attenuation of the clinical symptoms of Alzheimer’s dementia and reduction in DNA damage to a higher extent than the native non-conjugated AP [150].

The oral administration of EA-loaded PLC-NPs (EA-NPs), which proved to have high EE% and DL%, produced an EA plasma concentration of 3.6-fold higher than that produced by administering free EA [153].

Several structurally different eco-friendly soybean-oil-based cationic polyurethanes (PURs) were prepared to develop edible food coatings with antimicrobial properties toward a panel of bacterial pathogens, including *Listeria monocytogenes* NADC 2045, *Salmonella typhimurium* ATCC 13311, and *S. minnesota* R613. Tested against the same strains of wild-type, the PURs-based NPs exhibited better antibacterial activity on the Gram-positive *L. monocytogenes* than on the Gram-negative *S. minnesota* and excellent activity against *S. minnesota* R613 [154].

With the aim of ameliorating their performances, several EOs and their constituents have been subjected to modifications by nanotechnology and converted into NP formulations for improving their antimicrobial activity, thus allowing their exploitation to extend food shelf life and to minimize the growth of foodborne pathogens.

#### 2.1.3. Nanoparticle Characterization Techniques

In nanotechnology, it is crucial to find the right characterization techniques with optimum performances for studying the NPs characteristics. Many techniques exist for nanoparticle/nanomaterial characterization. Atomic force microscopy; dynamic light scattering; electron microscopy, including scanning electron microscopy (SEM) and transmission electron microscopy (TEM); FTIR spectroscopy; UV–visible spectroscopy; X-ray diffraction; and X-ray photoelectron spectroscopy are among the most adopted techniques. Table 2 collects the most common NP entities that should be investigated and characterized and a list of the techniques that are available for this scope, reported using their well-known acronyms for space reasons. Readers not expert in this field, who are interested, can still find the corresponding specifications online. Table 3 reports instead the techniques most frequently used, with some more detailed information, and the advantages and drawbacks related to each one.

#### 2.1.4. More In-Depth About Nanotechnology Applications to Nutraceuticals (Nuts) and Phytochemicals (Phys): In Vivo Experimental Advances

To exploit phytochemicals as health enhancers, researchers extensively engineered nanomaterials and resorted to nanotechnology and nanostructures with dimensions of nanometers (nms). The formulation of Phys using NPs has allowed their controlled and targeted release, which is essential for effective administration [79]. Controlled nano delivery translates into a higher concentration at the target, thus allowing a reduction in the overall administered dose and consequently systemic toxicity [1]. Both internal and external factors, such as pH, temperature, ultrasound or magnetic field application, light incidence, and the type and physicochemical features of NPs, as well as the chemical structure and the physicochemical features of the bioactive compounds themselves, can control their specific release [1].

Stimuli-sensitive nano-capsules containing a bioactive derivative of paclitaxel and possessing an oil *core* showed the capability to improve the anticancer effects of the encapsulated compound taken by oral administration, thanks to targeted delivery and controlled long-term release [86]. The improved effects allowed a decrease in the dosage and the administration frequency, thus improving patient compliance [86]. Starting from biocompatible pH-dependent polyelectrolytes, nontoxic nanocarriers with high permeability were designed [86].

The layer-by-layer self-assembly of pH-sensitive building blocks proved to be a promising approach to obtaining Phys-based biomaterials with customized properties, which were successfully applied as stimuli-responsive nanocarriers [160]. The encapsulation of bioactive compounds contained in food in properly functionalized NPs permitted increased cellular uptake and slower drug release, thus improving their bioactivity and contributing to sustained therapy [60].

According to a not-so-recent but relevant research paper, only up to the year 2019, while liposomes were the most studied NPs for nano-manipulating Phys, nano-emulsions (NEs) were little considered as a nanotechnological approach, while nano-suspensions (NSs) were not even reported [161]. Additionally, NSs and NEs are usually produced using ingredients regarded as safe (GRAS) like liposomes, so they should be considered among the less toxic and the most suitable tactics for developing Phys-loaded NPs finalized to clinical application. Anyway, due to this statistic, a relevant review dedicated to these too-little-considered nanotechnologies has been published in 2023. Such a study could be of interest to readers particularly attracted by the topic [62].

The poor solubility, permeability, and negative pharmacokinetics of a series of nutraceuticals (Nuts) were enhanced by developing different nanosized delivery systems [162]. Table 4 and Table 5 summarize some examples of nuts nanotechnologically formulated using different NPs and methods, associated with their activity as demonstrated by in vivo experiments and/or structural characteristics.

In addition to enhancing the bioavailability of Phys and Nuts, using different techniques, nanotechnology was and is used in the food sector to prepare NPs finalized to act as color additives, flavorings, and preservatives, as well as to prepare improved food packaging, with the aim to enhance food shelf life, taste, and appearance [79].

#### 2.1.5. More In-Depth About Nanotechnology Applications in the Food-Packaging (FP) Industry

In order to improve the mandatory properties of traditional materials for FP, which have been listed in studies by Kuswandy, including Kuswandi and Moradi, 2019 [206], and more recently reviewed by Alfei et al. in 2020 [68], nanotechnology is nowadays intensively studied, also for application in the FP industry. It has been demonstrated that the nanoencapsulation of bioactive natural compounds, by using particles with diameters ranging from 1 to 100 nm, leads to a remarkable increase in their solubility and stability, as well as to a decrease in their inactivation rate, thus offering the possibility of preparing better-performing food packaging exploitable as preservative agents in comparison to conventional ones [207]. The inclusion in food packaging of NPs with intrinsic antioxidant, antimicrobial, and antifungal properties or capable of releasing antioxidants, antimicrobial, preservative APs, flavors or enzymes, nutraceuticals, and/or phytochemicals previously entrapped can allow further improvements, including longer shelf life and overall higher food quality [208].

Collectively, the use of nanomaterials has improved FP both in physical and in biochemical characteristics [206,209].

Table 6 reports the main advanced FP types achieved using functional nanomaterials [68].

Note that both natural and synthetic polymers have been employed in the past to produce conventional FPs. Natural biopolymers include lipids-, polysaccharides-, and protein-based polymers like those previously discussed but obtained by the action of living organisms. They are completely degradable, while synthetic ones comprise both petroleum-based plastics and eco-friendly bio-based polymers, which can be in turn degradable and non-degradable materials. Degradable bio-based polymers and natural degradable biopolymers allow for a reduction in the levels of environmental pollution but lack suitable mechanical properties. Resorting to nanotechnology, both eco-friendly bio-based and petroleum-based nanocomposites have been prepared, which allow the development of nanomaterial-based physically improved FP. Anyway, to meet the increasing demand for cleaner, more economical, more versatile, and more sustainable packaging solutions, different eco-friendly materials are being searched [210]. Among these, lignin is emerging as a multifaceted, promising candidate for sustainable active food packaging applications. Specifically, lignin is the second-most abundant plant-based polymer on Earth, constituting approximately 20–35% (by dry mass) of lignocellulosic biomass, depending on the source [211]. Lignocellulosic biomass is the most abundant type of biomass on Earth, with annual global availability estimated at 181 billion tons [211,212]. Lignin results in being a low-cost, renewable, and biocompatible resource, rich in polyphenols that provide excellent antioxidant and antibacterial activity and protection against ultraviolet radiation [213]. For these reasons, lignin has been extensively employed as a multifunctional additive in food packaging polymers to prepare flexible films and coating formulations [210]. Recently, by formulating lignin as NPs, the efficacy of lignin as a multifunctional filler in food packaging has been remarkably improved [210]. Moreover, lignin hydrogels are reported to be suitable as “per se” food packaging or coating materials [214,215]. Anyway, more in-depth research on lignin macromolecules is needed to fill some residual knowledge gaps, mainly concerning ambiguities in their exact structure, mainly depending on the different isolation methods. Further studies are also necessary to unveil the exact impact of lignin structure on the physicochemical and functional properties of food packaging.

## 3. Our Dealing with Nanotechnology Applications: Last 15 Years Studies

Table 7 collects the main details about the nanosized materials developed in our laboratories in the years 2009–2024, while an additional twelve review articles we have published, concerning the most relevant nanomaterials for specific applications such as improvement in solubility and bioavailability of natural and/or synthetic APs, food sectors, food packaging, etc., were developed by other eminent scientists in the same years [60,61,62,63,64,65,66,67,68,69].

Since Table 5 appeared exhaustive in the description of the main structural and physicochemical characteristics and pharmacological effects of nanomaterials prepared by us, we did not dedicate further time to additional dissertations on them and preferred to drive readers attention towards a thorny and still foggy topic, such as the possible toxicity of nanoparticles.

## 4. Nanotoxicology

Nowadays, it is universally recognized that the application of nanotechnology can allow one to achieve many nonpareil advantages, regardless of the sector of application. Concerning APs, their nanomanipulation has permitted the efficient administration of not soluble and not bioavailable bioactive compounds and the preparation of unconventional food-related therapeutics. The protective action of nanoencapsulations of unstable APs has provided bioactive NPs capable of undergoing the strong conditions of processes necessary to enrich food with them, thus achieving functional foods (FFs), food supplements (FSs), as well as manufacturing active food packaging (FP) or preservative additives.

Many nanocomposites are still at the laboratory stage, but several products have already been approved by EFSA and by the Member States and the European Commission [218]. Nanotechnology application to APs, food, and beverages has had an exponential growth over the past 15 years, and due to the obtained advantages, the presence in the market of APs and foods nanotechnologically manipulated is destined to increase further. Anyway, a major problem remains to be solved, which concerns the poor knowledge about the possible risky effects on the health of humans, animals, and the environment, which can derive from ingestion and massive exposure to NPs and from the possible migration of NPs from FP into foodstuff [218]. In this context, the amplification of the development of nanomaterial-based food-related products is a topic debated incessantly among researchers with contrasting opinions, thus spreading concern and prejudice both among producers in various sectors and among consumers [219].

### The Possible Migration of NPs from FP to Food and Toxicity of Ingesting Them

The migration rate of NPs from nanocomposites used to manufacture innovative food packaging (FP) into food or food simulants has been measured using European and U.S. (FDA) standard migration tests. Anyway, numerous issues have emerged that complicated the determination and interpretation of NPs migration studies and related results [218]. Commonly, “migration into foodstuff” indicates the process by which the constituents initially present in the package, possibly including NPs, are liberated into the food or beverage packaged [68]. Limits of safety and the list of authorized substances for manufacturing polymeric food-contact nanomaterials have been established by the European Regulation (European Commission, 2011) [218]. Data on migration are available mainly for inorganic NPs, thus evidencing very restricted knowledge on the question. They mainly include nanoclay, titanium nitride, nano-silver, silanated silicon dioxide, titanium oxide, zinc oxide, and iron oxide NPs. Collectively, concerning these NPs, the reported findings have established that the migration of NPs from FP would be low and slow [68]. Additionally, the migration rate of a system increases when NPs size and polymer dynamic viscosity decrease [220].

The practice of inserting NPs in the FP materials is still in its infancy; therefore, only a few studies are available in the literature, and current information on their possible migration and on their toxicity upon exposure is limited. Further fundamental studies on toxicity, ecotoxicity, migration tendency, and the risk of the intake of nanocomposite materials are needed to authorize massive application of nanomaterials in the FP field [221]. What is the actual behavior of NPs once inserted in packaging and in contact with packaged foodstuff? What are the eventual mechanisms involved in the migration, and how can the diffusion process change the size and morphology of nanomaterials?

With the aim of answering these questions, a standardized food model (SFM) for evaluating the toxicity and fate of NPs migrated in a food matrix after ingestion was proposed for the first time by Zhang et al. in the year 2019, and its efficacy was assessed by examining the impact of the food matrix on the toxicity of TiO_2_ NPs [222]. SFM is an oil-in-water emulsion usable both in wet and in dry forms. Using a simulated GIT model, it was observed that all the SFM food components were well digested, and that the potential toxicity of TiO_2_ NPs was reduced in the presence of the SFM, underlining the importance of food matrix effects on the actual toxicity of NPs [222]. Table 8 collects the results achieved by evaluations made on some NPs, while Table 9 reports detailed migration results expressed wt/volume or as wt/wt of numerous inorganic NPs from different food contact materials (FCMs) into food or food simulants [218].

The overall migration of nanoclay/starch nanoparticles into vegetables was in conformity with European directives, thus establishing that these materials are suitable for utilization in the FP sector [68]. Similarly, the migration of PLA/laurate LDH-C12-modified NPs used to reduce the gas permeability of a packaging in a modified atmosphere for the conservation of meat resulted largely below the total legislative migration limits established for all materials [223]. Experimental results and theoretical considerations about the migration of two types of CNTs/LDPE/PS NPs in different food simulants established that such NPs do not migrate from the polymer matrix into food [224]. Even if a commercially available FP improved with AgNPs and intended to package chicken meatballs, under common domestic storage conditions, did not demonstrate significant antibacterial effects, it evidenced a migration rate encouragingly slow [225]. AgNPs did not migrate from AgNPs/PEF into chicken breast or distilled water after 168 and 72 h, respectively [226]. Inductively coupled plasma mass spectrometry (ICP-MS) analytical technique evidenced that migration of AgNPs from AgNPs/PE packaging films into an acidic food simulant (3% acetic acid) was promoted by the presence of organic additives, such as Irganox 1076, Irgafos 168, Chimassorb 944, Tinuvin 622, UV-531, and UV-P, and by Ag oxidation endorsed by high humidity and temperature treatment [227].

The evaluation of the degree of migration of NPs from active/smart/mechanically improved FP into foods and into the environment should be associated with an all-round knowledge of the possible toxic outcomes that the NPs eventually could have on humans and animals when ingested, as well as on the environment. NPs are not normally eaten and metabolized by humans and animal species, and, paradoxically, even if by adding NPs in FP, new advantageous opportunities can be achieved, such as ingesting higher-quality and safer aliments, new risks to human health and the environment can occur [218]. In this regard, the overall risks potentially associated with the intake of food containing NPs are still unclear. Currently, we know that smaller particles are usually absorbed easier and faster and are more promptly distributed into the organs. Here, NPs can damage cells and tissues by reactive oxygen species (ROSs) generation or by other types of direct or indirect toxicity [228]. As represented in Figure 6, the toxicological effects of NPs and their mechanisms have been evaluated in vitro and in vivo [229]. For evaluations in vitro, human and/or rodent cell lines from intestine, liver, lung, and skin, and plant cells, have been used. The cytotoxicity tests comprised the lactate dehydrogenase (LHD) release assay, live-dead assay, cell counting, Alamar blue assay, neutral red uptake, the evaluation of protein content, and trypan blue dye exclusion test. Specific mechanisms are generally studied by observing the changes in different biomarkers, such as ROS production, glutathione (GSH) levels, inflammation response, DNA damage, and cell death. In contrast, for assessing potential genotoxicity, the Comet assay, Ames test, and micronucleus essay have been adopted.

In vivo evaluations used mainly rodents and macro-toxic or histopathological effects; following the sub-chronic/chronic exposure to different kinds of NPs was assessed [229]. Unfortunately, the results from both in vitro and in vivo analyses are often contrasting and questionable. As an example, findings on nano organoclay have established that their toxicological evaluation, case by case, should be performed [229]. In addition, since the average amount of clay added as reinforcement of polymers should be around 5 wt%, it is important that the concentrations tested would be pertinent with a real oral exposure scenario. Collectively, more toxicity data from studies on an increasing number of nanocomposites are necessary for a more reliable evaluation of nanotoxicology and exposure estimation [230].

Table 10 collects the results from in vitro studies we have reviewed. Toxicity depended mainly on concentrations, type of cation of metal NPs, and especially morphology of NPs (nano wires, nano roads, nano spheres, etc.). Similarly, Table 11 reports the results from in vivo evaluations of different NPs with different morphologies on different animal models.

Although the potential toxicity by ingestion of Ag NPs is still debated, it has been found that the cytotoxicity reported in Table 8 (first row) can be nullified by the addition of carboxymethylcellulose (CMC) to the colloidal solutions of Ag NPs, while the genotoxic effects of Ag NP dispersions were observed at a concentration of 12.4 ppm [231]. One of the most accredited mechanisms by which SPM iron oxide NPs, also referred to as USPIO or SPION (USPIO = ultrasmall superparamagnetic iron oxide; SPION = superparamagnetic iron oxide nanoparticles), can induce cytotoxicity consists in causing an aberrant increase in ROS. By crossing the mitochondrial membrane, the free iron in the form of ferrous ions (Fe^2+^) can react with hydrogen peroxide and oxygen produced by the mitochondria to produce highly reactive hydroxyl radicals and ferric ions (Fe^3+^) via the Fenton reaction. Hydroxyl radicals generated could indirectly damage DNA, cause peroxidation of proteins and lipids, and generate inflammation [233]. Physiological doses of SiO_2_ NPs were tested in an in vitro model to assess their effects on gastrointestinal function and health, evidencing damage to the brush border membrane and both acute and chronic adverse effects in gastrointestinal tract (GIT) cells [243]. Furthermore, the results of an in vitro study have demonstrated that ZnO_2_ NPs may cause a decrement in the transport of Fe and glucose and affect the microvilli of the intestinal cells [244]. Anyway, the toxicity of ZnO_2_ NPs decreased when a surface modification by a silica coating was performed [245] and was proposed as a possible solution to broaden the applications of ZnO_2_ NPs as an antibacterial agent in FP.
nanomaterials-15-00265-t011_Table 11Table 11Results from in vivo studies on possible toxic effects to different animal models of different NPs at different concentrations having different morphologies (nano wires, nano roads, nano spheres, etc.).NPsConcentration(s)OrganismCytotoxicity/Genotoxicity/AssaysFindingsRefs.Ag NWs 100 nm4 μg/cm^2^*D. magna*Toxicity varied as a function of AgNW dimension, coating, and solution chemistry↑ Toxic ↓ Toxic than Au^+^[246]Ag NRs5–15 μM*Allium cepa*Mitotic index, chromosomal aberrationsROS assays↑ ROS and chromosomal damage (15 μM) [247]Au NRs TAB-capped 46.4 nmPEG-capped 48.1nm0.1–10 μg/mL*A. cepa*OS, lipid peroxidation assay↑ Mitotic index and OS [248]Ni NWs 20 nmNi NSs < 50 nm0.016–10 mM*D. melanogaster*Insignificant toxic effectsNo toxic or mutagenic impacts[249]Ti NRs <100 nmTi NWs <10 nmTi NSs <25 nm0.01–10 mM*D. melanogaster*Viability, internalization, intracellular ROS production, genotoxicity (comet assay)ROS and DNA damage (10 mM)Dose–effect in hemocytes[250]Ti NWs 14–95 m^2^/g10 μg/mL5-week-old ICR miceToxicity depended on the surface area of TiO_2_ NWs↑ Th2-type inflammatory cytokines ↑ Interleukin (IL)-1↑ Tumor necrosis factor-alpha (TNFα)↑ IL-6[251]BNNTs 4.56 nm0.01–10 mM*D. melanogaster*Non-significant toxic effects.BNNTs ↓ genotoxic effects of K_2_CrO_7_
↓ Intracellular levels of ROS[252]CNTs
Mice/ratInjected into the animal’s peritoneal cavity Peritoneal mesothelioma[253]GaP NWs 80 nm* 10 NWs nL^−1^
** 50 μL6 × 10^7^NWs mL^−1^*D. melanogaster*Not significantly affected life span or somatic mutation rateNot taken up into *Drosophila* tissuesNo measurable immune response No changes in genome-wide gene expression[254]GaP NWs 40 nm6.2 × 10^10^ NWs/L*D. magna*No mortality was observedPenetration of biological barriers governed by the NW diameter.[255]CdS NRs 30–50 nm 1000–10,000 mg/kgKunming mice (17–22 g/mouse)Apparent toxic effectsOS and DNA damage[256]USPION.R.HumansPhase III clinical trialUrticaria, diarrhea, nausea[257]SPION1.0 mg Fe/mouse per day for 15 daysMiceTrigger skin cancer↑ Stage-I, stage-II skin tumor [258]PAMNPs 8.5 nm0.6–1.6 × 10^10^ Ps/mLSwiss miceTime- and dose-dependent toxicity↑ Micronucleus frequency[259]SiO_2_ NPs 10 nm*** 2 mg/kg RatsAlterations in morphometry, biochemistry, hematology, liver tissues, and the expression of drug-metabolizing enzyme genes↑ Alkaline phosphatase, LDH, low-density lipids, procalcitonin, aspartate aminotransferase, alanine aminotransferase↑ K, P, and Fe concentrations[260]NRs = nano roads; NW = nano wires; OS = oxidative stress; NSs = nano spheres; ↓ = reduced, decreased, low, lower; ↑ improved, increased, high, higher, highly; USPIO = ultrasmall superparamagnetic iron oxide; SPION = superparamagnetic iron oxide nanoparticles; N.R. = not reported; TBIS = total body iron stores; PAMNPs = magnetite NPs coated with poly aspartic acid; BNNTs = boron nitride nanotubes; CNTs = carbon nanotubes; *, **, ***: 20, 35, and 50 repeated injections.

As evidenced in Table 9, the number of in vivo studies performed in humans is very limited. One investigation found that Ferumoxtran-10, an iron-dextran compound coated with ultrasmall superparamagnetic iron oxide (USPIO) used in diagnostics, induced only side effects such as urticaria, diarrhea, and nausea, all of which were mild and short in duration [257]. In vivo and in vitro studies using rats or mice lung, glia, and breast cells evidenced that the iron released from superparamagnetic iron oxide nanoparticles (SPIONs) could lead to an iron overload in specific targeted organs or tissues with toxic implications such as an imbalance in iron homeostasis and can cause aberrant cellular responses, including cytotoxicity, DNA damage, OS, epigenetic events, and inflammatory processes [261,262,263,264]. The deleterious cellular disruption in the form of DNA damage caused by iron accumulation may initiate carcinogenesis due to a hyper-generation of ROS that can potentiate direct damage to DNA, proteins, and lipid peroxidation [233]. Iron overload following intra-muscular injections of an iron–dextran complex has been associated with spindle cell sarcoma and pleomorphic sarcoma in rats [258]. It has been found that iron overload is associated with the production of hydroxyl radicals in rats, which react with membrane lipids, giving rise to breakdown products, including malondialdehyde (MDA) and 4-hydroxy-2-nonenal (HNE), both of which can bind to DNA and are mutagenic [233]. Furthermore, an increased number of DNA breaks have been demonstrated in rats subjected to dietary iron overload, while oxidative damage to DNA has been observed in mice administered with iron–dextran complex [265]. An in vivo study on Swiss mice using poly aspartic acid-coated magnetite NPs demonstrated a time- and dose-dependent increase in micronucleus frequency [259]. CNTs caused peritoneal mesothelioma when injected into the rats or mice’s peritoneal cavity [253]. Different mineral clays were assayed in vitro using human umbilical vein endothelial cells (HUVECs), and their possible mutagenicity was assessed by Ames test. While unmodified Cloisite^®^ Na^+^ did not show any cytotoxic or mutagenic effect, Cloisite^®^130B) showed both toxic and mutagenic effects [229].

Although both in vitro and in vivo studies have established that TiO_2_ NPs accumulate in the tissues of mammals and are eliminated very slowly, the results on both their accumulation and toxicity are conflicting. More reliable in vivo toxicokinetic data are needed to provide conclusions concerning the risk of TiO_2_ NPs oral exposure [218].

Some in vivo studies on SiO_2_ NPs showed that they may cause cytotoxicity and ROS generation and may accumulate in the liver, causing hepatotoxicity, evidenced by alterations in morphometry, biochemistry, hematology, liver tissues, and the expression of drug-metabolizing enzyme genes [260]. In contrast, other studies, performed by administering SiO_2_NPs as well as Fe NPs to both female and male rats over a 13-week period, reported no accumulation or toxicity [218].

Antimicrobial ZnO_2_ NPs and ZnO_2_ NPs can reach several organs through ingestion, inhalation, and parenteral routes. Their oral administration induced neurotoxicity and a proinflammatory response in rats and immunotoxicity in different ages of BALB/c mice [266,267].

## 5. Conclusions

The biomedical revolution is governed by nanotechnology, which aids in resolving several issues associated with most natural and/or synthetic APs, thus limiting their effectiveness as well as their actual applicability in vivo. Target treatment administration and maximized therapeutic effectiveness can be achieved, while side effects can be reduced by formulating APs using nanomaterials. Applying nanotechnology, vaccines, anticancer drugs, antimicrobial agents, and other nanomedicines can be engineered, while wearable, diagnostic, and imaging equipment can be realized. Combining standard medicines with nanoscale technologies, the blood–brain barrier (BBB) can be crossed intact, and nanodrugs can circulate inside the brain. This technology offers enormous potential markets and benefits to whole classes of current drugs. It is possible to develop tailored mechanisms for medication administration, new diagnostic methods, and nanoscale medical devices with high residence time. Technological advancements in the fields of nanoscience and nanotechnology have led to a remarkable innovation also in the food sector that could bring wide-ranging benefits to the whole food chain. Such innovations include the development of new tastes, textures, mouth sensations, and consistencies of food products. The reduction in the amount of fat and certain additives, such as salt, an enhancement in the absorption and bioavailability of nutrients and supplements, and the preservation of food quality and freshness represent other innovative challenges that are progressively solved by nanotechnological studies. The research for novel nanomaterial-based packaging solutions, allowing better traceability and security of food products in the supply chain, is incessantly in rapid expansion. Food packaging (FP) applications currently represent the largest portion of the nanofood market, following the nanosized and/or nano-encapsulated ingredients and additives for healthy food production. Although still new and scarce in Europe and other regions, a discrete amount of nanomaterial-improved food-contact materials (FCMs) and more functional food products containing nanosized ingredients and additives are already available in some countries. It is anyway rational to imagine that such products will be available on the global market in increasing numbers and variety in the coming years. Their expansion will depend mainly on the price, quality, and, above all, acceptance by consumers. Although nanotechnology applications for the food, healthy food, and biomedicine sectors have undoubtedly unlocked enormous opportunities for innovation and new developments, they have also opened new happenstances for ensuring safety and evidencing all the potential risks of this new technology, without highlighting only the benefits. Reliable strategies to better know the risks for humans, animals, and the environment, which can derive from a massive exposition to nanomaterials and to regulate them in a globally harmonized manner, are needed. Unfortunately, food laws in different countries may not conform to each other, and this fact is a challenge to the regulatory authorities. In this regard, it would be desirable that in due course, such issues could be addressed through the development of frameworks relating to key international trade agreements, such as those administered by the World Trade Organization. We hope that this new review can provide much-needed insights into the various aspects and issues relating to the new and exciting developments that nanotechnologies are offering to the food, medicine, and related sectors. According to scientists, the perspective lines of nanotechnology and of the production of nanomaterials were predicted to encompass four distinct generations of advancements. We are currently undergoing the first, or possibly the second, generation of nanomaterials. The first generation concerns the material science aiming at the enhancement of properties of materials obtainable by incorporating “passive nanostructures”, by coatings and/or using carbon nanotubes to strengthen plastics. The second generation adopts active nanostructures, which are bioactive. They could be capable of driving a drug at a specific target cell or organ due to specific proteins used to coat their surface. Advanced nano-systems such as nanorobotics represent the third-generation nanotechnology, while moving on to a molecular nano-system to control the growth of artificial organs could be the fourth generation of nanomaterials. The ‘safe-by-design’ nanomaterials is a concept currently under investigation by scientists and in development. The proposition consists of incorporating the safety assessment of nanoparticles into the design and innovation stage of nanomaterial’s development, rather than testing the safety of nanomaterials after they are put on the market. The aim of this project is to give companies more cost-effective risk management early in the process and/or product developments.

## Figures and Tables

**Figure 1 nanomaterials-15-00265-f001:**
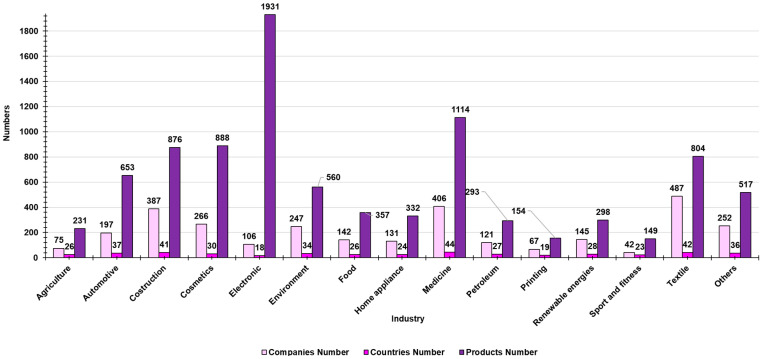
Industry-wise division of several companies, countries, and products using nanotechnology. Data concern a picture of June 2021, according to the Statnano website (STATNANO: Nano Science, Technology, and Industry Information, available online at https://statnano.com/).

**Figure 2 nanomaterials-15-00265-f002:**
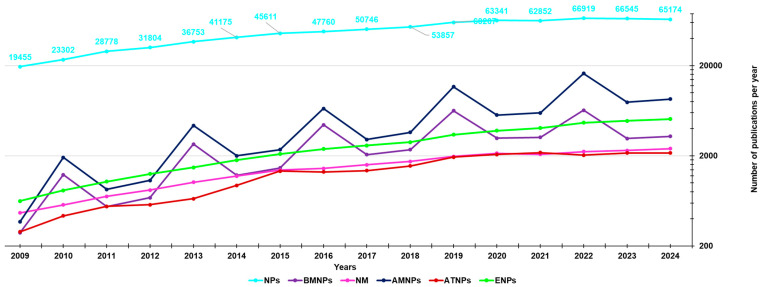
Number of publications found in the literature and published in the last fifteen years (2009–2024), achieved from a survey conducted using the Scopus database (https://www.scopus.com/, accessed on 6 November 2024). The values on the y-axis are in logarithmic scale to reduce the space between the light blue line and the others. Only experimental studies were considered intended as full articles, conference papers, and letters. They were found using the keyword “nanoparticles” (light blue line), using as keywords “antimicrobial AND nanoparticles” and then “antibacterial AND nanoparticles” (blue line), using “anticancer AND nanoparticles” and “antitumor AND nanoparticles”, and then “antitumor AND nanoparticles” (red line), using “biomedical AND nanoparticles” (purple lines), using “environment AND nanoparticles” (green line), and finally that obtained using “nanomedicine” (pink line).

**Figure 3 nanomaterials-15-00265-f003:**
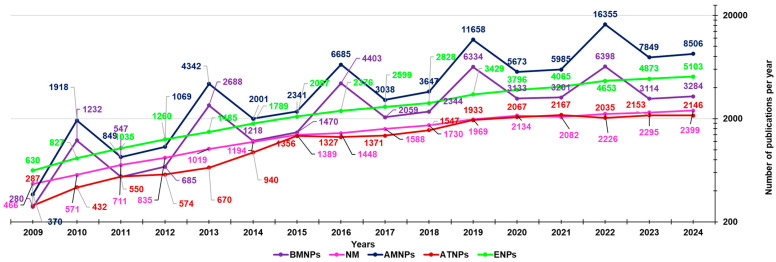
Number of publications found in the literature and published in the last fifteen years (2009–2024), achieved from a survey conducted using the Scopus database (https://www.scopus.com/, accessed on 6 November 2024). The values on the y-axis are on a logarithmic scale for a better distribution of lines within the graph. Only experimental studies were considered intended as full articles, conference papers, and letters. They were found using as keywords “antimicrobial AND nanoparticles”, “antibacterial AND nanoparticles” (blue line), “anticancer AND nanoparticles”, “antitumor AND nanoparticles”, and then “antitumor AND nanoparticles” (red line), using “biomedical AND nanoparticles” (purple lines), using “environment AND nanoparticles” (green line), and finally that obtained using “nanomedicine” (pink line).

**Figure 4 nanomaterials-15-00265-f004:**
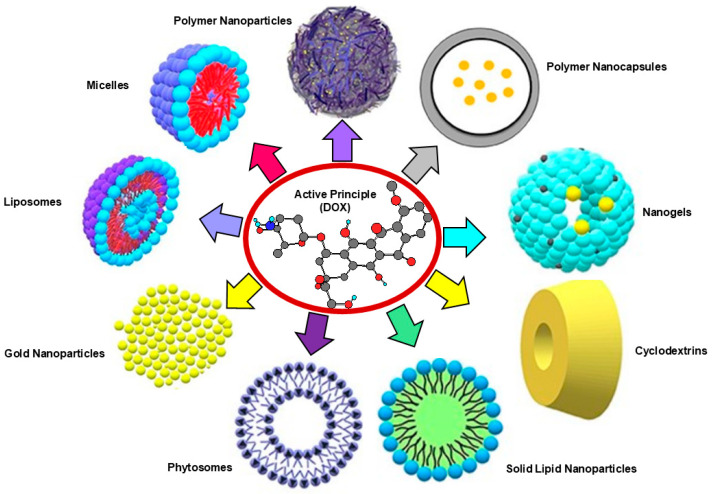
Possible nano-formulation routes reported in the literature for lipophilic antitumor APs, such as doxorubicin (DOX), to achieve nanosized dosage forms for more efficient anticancer therapy.

**Figure 5 nanomaterials-15-00265-f005:**
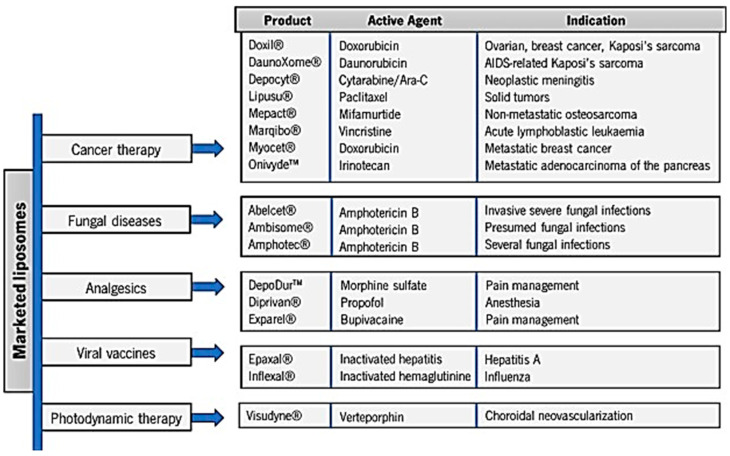
Marketed liposomal formulations. Reproduced from [143] with permission of Copyright Clearance Center’s RightsLink^®^ service.

**Figure 6 nanomaterials-15-00265-f006:**
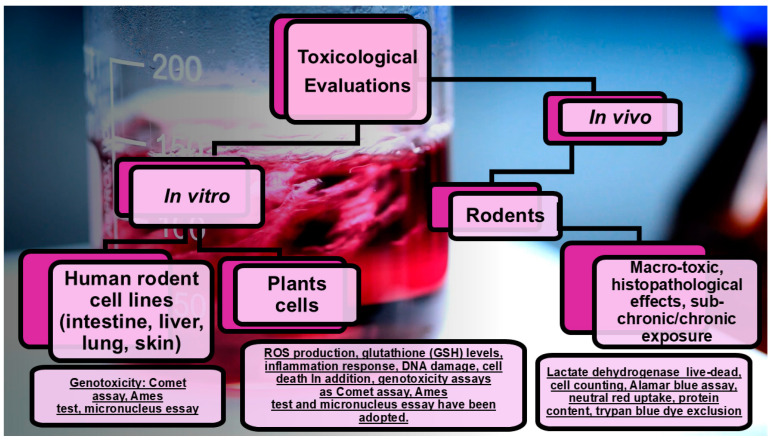
Developed in vitro and in vivo toxicological evaluations of NPs.

**Table 1 nanomaterials-15-00265-t001:** Most used nanomaterials engineered for encapsulating natural and synthetic APs for biomedical uses and health purposes.

**Method**				**Description**	**General Indications/Uses**	**Properties**	**Main APs**	**Refs.**
NSs				Colloidal dispersion of NPs (10–900 nm) in water Surfactants, co-surfactants, polymers *	To improve the solubility/bioavailability of both HAPs/LAPs	↑ Dispersibility, ↑ solubility Sustained, controlled, targeted delivery ↑ Stability ↑ Therapeutic effects in cells and tissues		[60]
							β-carotene	[88]
							Quercetin	[89]
							Acai fruits	[90]
							α-tocopherol	[91]
NEs				Kinetically stable liquid-in-liquid dispersions, with droplet sizes of 100–500 nm	↓ Particle size of HAPs/LAPs, H/L food additives ** Orally administrable drugs Protected drug delivery Suitable for food, cosmetics, pharmaceuticals Suitable for material synthesis	↑ Solubility/bioavailability Sustained, controlled, targeted delivery Extended half-life Obtained either by low-energy techniques or by high-energy techniques		[92]
							Turmeric	[60]
							Curcumin	[93]
							di-Benzoyl-methane	[60]
							Tannins Stilbene Flavonoids	[94]
							ECGC	[95]
							Lipids Carotenoids	[60]
							Pomegranate extracts	[60,96]
							LEO	[97]
SEDDSs		SMEDDSs 100–200 nm		Anhydrous nano-dispersions achieved by drying ^A^ an oil phase, surfactants, co-surfactants/co-solvents, and LAPs Powders will spontaneously arrange in colloidal NEs when merged with water or with GIT fluids by small agitation or by the digestive motility of the stomach and intestine	For orally delivering LAPs, food-grade chemicals, additives, and drugs For low therapeutic dose APs	↑ Oral bioavailability improvement Possibility of an easy scale-up ↑ DL% Allow delivering peptides and lipids without the risk of lipid digestion		[98,99]
		SNEDDSs < 50 nm					EGCG	[100]
SDDSs				↑ Soluble bioactive NPs with the AP physically entrapped or covalently linked (20–1000 nm) Nanocarriers can be made of PEG, PUR, PCL, PLGA, PVA, P2VP, PLA, PPO, Pluronics^®^, PGA, PAE, PLL, mPEG, PasP, PLH, PEI, PVP, PLLeu, DOCA, HPMC, PHB, PEO, PBLG, PS, PIHCA, PAH, and biocompatible polyester-based dendrimers	For delivering HAPs/LAPs, food-grade chemicals additives, drugs For low therapeutic dose APs	↑ Solubility, bioavailability, dispersion, and stability in GIT ↑ APs systemic spread, transportation through the endothelial cell layer ↑ Release at the target site Controlled microbiota metabolism ↑ APs’ bio-efficacy ↑ Cellular uptake Favorable drug release profile protracted in time		[101,102,103]
							Paclitaxel ^B^ Doxorubicin ^B^ mPEG-PLGA-Paclitaxel ^B^ PEGylated factor VII ^C^ Estradiol ^C^ PEGylated antibody fragment ^C^ Erythropoiesis stimulating agent ^C^ PEGylated IFNbeta-1a ^C^	[104]
							Dexamethasone Docetaxel Rifampin Genistein + paclitaxel + quercetin Hydrochlorothiazide Cisplatin Curcumin Diminazen aceturate Paclitaxel Folic acid siRNA + paclitaxel Docetaxel +siRNA-Bcl-2 Doxorubicin Lidocaine Cripofloxacin HCl Dexamethasone Insulin FITC-Dextran Levonorgestrel DNA	
OSNPs	LNPs		SLNPs	An external lipid monolayer with a solid-lipid core Spherical morphology (10–1000 nm) Surfactants/emulsifiers to stabilize Ideal fat/aqueous medium ratio 0.1/30.0 (*w*/*w*)	For delivering LAPs	Biocompatible	Domperidone	[105]
			LPs	Artificial vesicles achieved by mixing phospholipids + cholesterol Lipid bilayer enclosing an aqueous core	Immunological adjuvants and drug carriers	↑ EE% of APs with different polarities Preserve APs from enzyme activity and degrading agents Biodegradable, biologically inactive Non-antigenic, non-pyrogenic, no intrinsic toxicity, instability in plasma ^D^	Irinotecan Amphotericin B Verteporfin Morphine sulfate Bupivacaine Inactivated hepatitis A Inactivated hemagglutinin of influenza A and B	[60,106]
			n-MIC	Very slim, spherical lipid particles (10–400 nm)	n-MICs form in aqueous medium n-MIC can solubilize LAPs	↑ Bioavailability ↑ Systemic residence time Protect APs from early inactivation ↑ DL% and good stability	Paclitaxel Doxorubicin Curcumin Dextran/Doxorubicin Doxorubicin/SN-38 Podophyllotoxin LCA Doxorubicin/siPD-L1 β-Lapachone/camptothecin Doxorubicin/CD147 miR-34a mimic/volasertib (BI6727) siRNA siRNA/Doxorubicin Docetaxel Sorafenib Camptothecin Paclitaxel/siRNA Dexamethasone JQ1 Estradiol Adriamycin Doxorubicin/Fe_3_O_4_ NPs	[107]
			i-MIC		i-MICs form in oil medium i-MIC solubilizes HAPs			
			NIOs	Uncharged or charged lipid-based lamellar nanostructures Merge non-ionic ^E^, cationic ^F^, or anionic ^G^ surfactants + cholesterol Vesicles are osmotically active/stable	For ↑ oral bioavailability of APs with limited absorption	↓ Toxicity for cells *** Act as reservoir systems Provide controlled and sustained delivery	Tamoxifen Docetaxel Metformin Celecoxib Gemcitabine Ascorbic acid Geranium oil Curcumin Cisplatin, Epirubicin Folic acid Letrozole Cyclophosphamide, Farnesol Gingerol Doxorubicin Hyaluronic acid Morusin Melittin Paclitaxel 2,5-Diketopiperazine Carnosine Trastuzumab Mcl-1 Nioplex Nintedanib Artemisin Silibinin Sunitinib 5-Fluorouracil Oxaliplatin Saccharomyces Cerevisiae Lycopene Hippadine γ-Oryzanol Amygdalin Ozonated olive oil	[108]
	Pro NPs			Made of both animal ^I^ and vegetable proteins ^L^ through protein precipitation and cross-linking agents ^§^ De-solvating agents ^M^	For carrying several molecules	Simple manufacturing Compatible with the ↑ -pressure Emulsification processes ↑ Freeze–thaw stability Suitable for being transformed Biocompatibility ↑ Stability ↑ Permeation ability in vitro Sustained delivery of APs ↓ Toxicity for cells ^#^ ↑ Shelf life of APs ↑ Resistance of APs to acidic gastric pH	EGCG, GA Probiotics	[15,109,110,111,112]
	ONPs		CDs	Cyclic oligosaccharides consisting of six (α-CD), seven (β-CD), eight (γ-CD), or more glucopyranose units linked by α-(1, 4) bonds	For preparing FFs, FSs, IFT, and APs by the monomolecular inclusion complex technique To deliver different LAPs	↑ Hydrophilicity and water solubility of LAPs ↑ Chemical stability ↓ Early degradation and metabolism Can modify unpleasant tastes and flavors Realize a controlled release of LAPs	Linoleic acid Resveratrol Carotenoids Lycopene (Lyc) Hesperidin Olive leaf extracts 1 Quercetin Myricetin Kaempferol 3-Hydroxyflavone Morin Rutin Curcumin Ferulic acid Ellagic acid Amino acids Hydrolyzed soy pro 2	[60,69,113,114,115,116,117,118]
	PNPs			Prepared from natural hydrophilic polysaccharides Comprise polyelectrolytes (cationic, anionic, neutral saccharides) and non-polyelectrolytes	To deliver different APs To develop APs-load FFs and additives	↑ Solubility ↑ Controlled and target release ↑ Stability, ↑ food shelf life ↑ Cellular uptake	Olive leaf extract Gallic acid Caffeic acid Yerba mate ^H^ Caffeine Theobromine Saponins Polyphenols Probiotics Flavors Anthocyanins Procyanidins Ellagic acid Gliadin	[60,119,120,121,122,123,124,125,126,127,128,129,130]

* To stabilize the system; HAP = hydrophilic active principle; LAP = hydrophilic active principle; ** not high-melting APs; SEDDSs = self-emulsifying drug delivery systems; ^A^ = through proper procedures, including spray dry (SD) or freeze drying (FD); SNEDDSs = self-nanoemulsifying drug delivery systems; SMEDDSs = self-micro-emulsifying drug delivery systems; ECCG = epigallocatechin-3-gallate; LEO = lemongrass oil; PEG = polyethylene glycol; PUR = poly urethane; PCL = poly capro-lactone; PLGA = poly lactic-co-glycolic acid; PVA = polyvinyl alcohol; P2VP = poly 2-vinyl pyridine; PLA = poly(lactic acid); PPO = poly(propylene oxide); Pluronics^®^ = PPO-PEO; PGA = poly(γ-L-glutamic); PAE = poly(L-phenylalanine ethyl ester); PLL = poly(L-Lysine); mPEG = methyl-PEG; PasP = poly(aspartamic acid); PLH = poly(L-histidine); PEI = poly(ethylene amine); PVP = poly(N-vinylpyrrolidone); PLLeu = poly(L-Leucine); DOCA = deoxycholic acid; HPMC = hydroxy propyl methyl cellulose; PHB = poly(hydroxy butyrate); PEO = poly(ethylene oxide); PBLG = poly(γ-benzyl-L-glutamate); PS = phosphatidylserine; PIHCA = poly(isohexyl-cyanoacrylate); PAH = poly(allylamine hydrochloride); ^B^ = in clinical trial; ^C^ = approved; SDDSs = solid drug delivery systems; OSNPs = organic solid nanoparticles; LNPs = non-synthetic lipid-based NPs; ProNPs = protein-based polymeric NPs; ONPs = oligosaccharides-based NPs; PNPs = polysaccharides-based NPs; SLNPs = solid-lipid nanoparticles; LPs = liposomes; n-MICs = normal micelles; i-MIC = inverse micelles; NIOs = niosomes; ^D^ = sterically stabilized LPs have been developed; ^E^ = alkyl or di-alkyl polyglycerol ethers; ^F^ = stearyl pyridinium salts; ^G^ = di-acetyl phosphate; *** for non-charged niosomes; CDs = cyclodextrins; FFs = functional foods; FSs = food supplements; IFT = innovative food-related therapeutics; ^H^ = *Ilex paraguariensis*; ^I^ = gelatin, collagen, albumin, casein, and silk protein; ^L^ = zein, gliadin, and soy protein; ^M^ = alcohol or acetone; § = chemical, ionic, thermal, and enzymatic (8% glutaraldehyde aqueous solution or calcium phosphate); ↓ = reduced, decreased, low, lower; ↑ improved, increased, high, higher; ^#^ for not cationic materials.

**Table 2 nanomaterials-15-00265-t002:** Parameters needed to be determined and the acronyms of the corresponding characterization techniques.

Entity Characterized	Characterization Techniques Suitable
Size (structural properties)	TEM, XRD, DLS, NTA, SAXS, HRTEM, SEM, AFM, EXAFS, FMR, DCS, ICP-MS, UV-Vis, MALDI, NMR, TRPS, EPLS, magnetic susceptibility
Shape	TEM, HRTEM, AFM, EPLS, FMR, 3D tomography
Elemental-chemical composition	XRD, XPS, ICP-MS, ICP-OES, SEM-EDX, NMR, MFM, LEIS
Crystal structure	XRD, EXAFS, HRTEM, electron diffraction, STEM
Size distribution	DCS, DLS, SAXS, NTA, ICP-MS, FMR, superparamagnetic relaxometry, DTA, TRPS, SEM
Chemical state—oxidation state	XAS, EELS, XPS, Mössbauer
Growth kinetics	SAXS, NMR, TEM, cryo-TEM, liquid-TEM
Ligand binding/composition/density/arrangement/mass, surface composition	XPS, FTIR, NMR, SIMS, FMR, TGA, SANS
Surface area, specific surface area	BET, liquid NMR
Surface charge	Zeta potential, EPM
Concentration	ICP-MS, UV-Vis, RMM-MEMS, PTA, DCS, TRPS
Agglomeration state	Zeta potential, DLS, DCS, UV-Vis, SEM, Cryo-TEM, TEM
Density	DCS, RMM-MEMS
Single particle properties	Sp-ICP-MS, MFM, HRTEM, liquid TEM
3D visualization	3D tomography, AFM, SEM
Dispersion of NP in matrices/supports	SEM, AFM, TEM
Structural defects	HRTEM, EBSD
Detection of NPs	TEM, SEM, STEM, EBSD, magnetic susceptibility
Optical properties	UV-Vis-NIR, PL, EELS-STEM
Magnetic properties	SQUID, VSM, Mössbauer, MFM, FMR, XMCD, magnetic susceptibility

**Table 3 nanomaterials-15-00265-t003:** Nanoparticle characterization techniques.

Technique		Useful Information	Advantages	Disadvantages	Refs
AFM		For particle size and morphology assessment 3D images are obtained (AF topography) The preparation methods are drop-deposition, adsorption deposition, and ultracentrifugation	Possible to analyze samples under moist conditions or even in liquids, which affords minimum perturbation	Lateral dimensions are greatly overestimated in liquids	[155]
DLS		For determining the size and PDI of NPs Gives an intensity-weighted correlation function that can be converted to an intensity-weighted (z-average) diffusion coefficient	Rapid, simple Readily available equipment ↓ Perturbation of sample	Unreliable interpretation * Critical review of the data obtained	[155]
EM	SEM	For the particle size and morphology assessment The interaction of the beam with the particle surface is scanned over the sample and measured as secondary electrons backscattered electrons or X-ray photons Due to the high depth of field in SEM, a three-dimensional appearance can be obtained	Reliable, highly repeatable results, with unprecedented resolution	The sample needs to be conductively coated with gold or graphite and maintained under ultrahigh vacuum, which can influence the results	[155,156]
	TEM	Utilizes energetic electrons to provide morphologic, compositional, and crystallographic information on samples Two-dimensional black and white images Maximum potential magnification of 1 nanometer	High-resolution Wide range of educational, science and industry applications Powerful	Very expensive Laborious sample preparation Potential artifacts Required special training Electron-transparent samples, able to tolerate the vacuum chamber and small enough to fit in the chamber, are needed Special housing and maintenance required Images are black and white	[157]
FTIR		Provides information on molecular structure, molecular interactions, and supramolecular assemblies Characteristic bands represent “fingerprints” of nanomaterial-(bio)molecule and/or organic molecule conjugation Allow monitoring of lipid content in nanosized liposomes designed for drug delivery To study the conformation of functional molecules covalently grafted onto carbon nanotubes, silica-based calcium di-methylene-tetraamine-pentakis (methylene phosphonate) NPs Suitable for metal NPs, dendrimers, and functionalized dendrimers	Non-destructive, highly reproducible, sensitive Employed for both qualitative and quantitative characterization	Inability to determine the complete chemical structure of a compound Difficulty in identifying complex samples Sensitivity to water, Sample preparation requirements Spectral interference	[158,159]
UV–Vis		For quantitation of particle concentrations if the optical constants of the particles are known	Rapid, simple Readily available equipment	Possible interference by background absorption for nanoparticle quantitation in aquatic systems Affected by particle size	[155]
XRD		Information on the crystalline structure, phase nature, lattice parameters, and crystalline grain size A method of measuring inter-particle spacings resulting from interference between waves reflecting from different crystal planes Can be used to distinguish between the anatase and rutile and amorphous phases of TiO_2_ NPs	Produce statistically representative, volume-averaged values The particle composition can be established	Not ideal for amorphous materials Peaks too broad for particles < 3 nm A dry sample needs to be prepared as a thin film	[155,159]
XRPS		For surface chemical analysis Its underlying physical principle is the photoelectric effect Powerful quantitative technique, useful to elucidate the electronic structure, elemental composition, and oxidation states of elements in a material It can also analyze the ligand exchange interactions and surface functionalization of NPs, as well as core/shell structures It operates under ultra-high vacuum conditions	Probes the composition of the material along the direction of the electron beam Direct and accurate empirical method to convert the XPS intensities into overlayer thicknesses. Provides the depth information, like the size of NPs Conservative	Preparation of samples (i.e., dry solid form is required without contamination) Difficult interpretation of data.	[155]

AFM = atomic force microscopy; DLS = dynamic light scattering, FTIR = FTIR spectroscopy, UV-Vis = UV–visible spectroscopy, XRD = X-ray diffraction, XRPS = X-ray photoelectron spectroscopy; EM = electron microscopy; * especially for polydisperse systems; PDI = polydispersity index; SEM = scanning electron microscopy; TEM = transmission electron microscopy.

**Table 4 nanomaterials-15-00265-t004:** In vivo experiments with nut-loaded delivery systems.

NPs	Nuts	Bioactivities	(nm)	Animal	Refs.
*Phospholipid-based delivery systems*					
Liposome	(+)-Catechin	Antioxidant, neuroprotective	35–70	Wistar albino rats	[163]
Liposome	Curcumin	Anti-HIV, antitumor, antioxidant Anti-inflammatory	263	Sprague-Dawley rats	[164]
HL	Silymarin	Hepatoprotective	660	albino rats	[165]
PLS	Silymarin		196	Beagle dogs	[166]
PPC	*Ginkgo biloba **	Polydispersity index platelet aggregation Radical scavenger, antioxidant, protection of CNS	N/A	Sprague-Dawley rats	[167]
PPC	Curcumin	Antioxidant, anti-inflammatory Anticarcinogenic, antibacterial ↓ Cholesterol, antitumor, antispasmodic Wound healing, anticoagulant, hepatoprotective		Wistar albino rats	[168]
PPC	Evodiamine	Antitumor, anti-inflammatory, anti-obesity Antinociceptive, thermoregulatory		Sprague-Dawley rats	[169]
PPC	Silybin	Hepatoprotective		Rats	[170]
PPC	Boswellic acid	Anti-inflammatory, hepatoprotective ↓ 5-Lipoxygenase		Rats	[171]
PPC	Silybin	Hepatoprotective, antioxidant		Dogs	[172]
*Emulsion-based delivery systems*					
NEs	DBM	Anticancer activities, anti-proliferation	70	Sprague-Dawley rats	[173]
NEs	α-tocopherol	Antioxidant, neuroprotective	85	Wistar rats	[174]
MEs	Berberine	Antibacteria, antitumor, anti-diabetes ↑ Cerebral ischemia	24	Sprague-Dawley rats	[175]
MEs	Puerarin	Cardiovascular diseases, antioxidants, and anti-diabetes	40	Kunming mice	[176]
SLNPs	Camptothecin	Anticancer	197	C57BL/6J mice	[177]
SLNPs	Quercetin	Antioxidant, ↓ blood lipid, anticancer ↓ Platelet aggregation, anti-anemia Anti-inflammation, anti-anaphylaxis Dilate coronary arteries	155	Wistar rats	[178]
SLNPs	Triptolide	Immune-suppressive activity Anti-fertility, anti-neoplastic activity	116	Wistar rats	[179]
OG NEs	Curcumin	Anticancer, anti-inflammatory, antioxidant	218	CD-1 mice	[180]
SEDDSs	Curcumin	Anti-inflammatory, antioxidant, anticancer	85	Wistar rats	[181]
SEDDSs	*Ginkgo biloba* *	↓ Platelet aggregation, radical scavenging Antioxidant, protection of CNS	∼100	Dogs	[182]
SEDDSs	Wurenchun	Antihepatotoxic, hepatoprotective	240	Sprague-Dawley rats	[183]
SEDDSs	Baicalein	Anti-inflammatory, anticancer, antioxidant Antivirus, antiallergic	27–54	Sprague-Dawley rats	[184]
SEDDSs	ZTO	Hepatoprotective, ↓ tumor, antibacterial ↑ White blood cell, anti-thrombosis	182	Rabbits	[185]
SEDDSs	Oridonin	Antitumor, antibacterial, antioxidant Anti-inflammatory	24	SD rats	[186]
*Chemical modifications*					
PAc	EGCG	Antioxidant, anti-viral, anti-inflammatory Cardioprotective, neuroprotective Anticancer	N/A	CF-1 mice	[187]
3,5,4′-TAR **	Resveratrol	Anti-cardiovascular disease, antioxidant Anti-inflammatory, antitumor		Rats	[188]
QC-12 **	Quercetin	Antioxidant, ↓ blood lipid, anti-anemia ↓ Platelet aggregation, anticancer Anti-inflammation, anti-anaphylaxis Dilate coronary arteries		Human	[189]
HAA	Tricin	Antioxidant, anti-viral, anti-inflammatory Antihistamine, anticancer		SD rats	[190]
*Other delivery methods*					
Ch NPs	EGCG	Antioxidant, anti-viral, anti-inflammatory Cardioprotective, neuroprotective, anticancer	440	Swiss outbred mice	[191]
Ch NPs	Curcumin	Antioxidant, anti-inflammatory, anti-proliferative, anti-angiogenic	178	Swiss mice	[166]
Naked NCs	Coenzyme Q10	Co-factor of the mitochondrial electron transport chain, antioxidant, cardioprotective neuroprotective	400, 700	Beagle dogs	[192]
NCs	Schisandrin B	Hepatoprotective, neuroprotective	45, 168	Sprague-Dawley rats	[193]

* Extracts; ** prodrug; HL = hybrid liposome; MEs = microemulsions; CNS = central nervous system; DBM = di-benzoyl methane; OG NEs = organo-gel nanoemulsions; PPC = phosphatidylcholine; SEDDSs = self-emulsifying drug delivery systems; SLNPs = solid-lipid nanoparticles; Ch NPs = chitosan nanoparticles; NPs = nanoparticles; 3,5,4′-TAR = 3,5,4′-tri-O-acetyl resveratrol; ZTO = Zedoary turmeric oil; HAA = hybrid with amino acids; NC = nanocrystals; EGCG = epigallocatechin gallate; PLSs = proliposome; PPCs = phospholipid complex; PAc = peracetylation.

**Table 5 nanomaterials-15-00265-t005:** In vivo experiments with a specific Nut nano formulated using different nanomaterials with enhanced activity.

Nuts	Technologies	Efficacy Evaluated	Models	Refs.
Curcumin	PPC	Antioxidant, hepatoprotective	CCl_4_-I liver OD in mice	[168]
	Nanoencapsulation	Chemopreventive	DENA-I liver cancer in rats	[194]
	NE	Anti-inflammation	TPA-I acute mouse ear edema	[195]
Quercetin	PPC	Antioxidant, hepatoprotective	CCl_4_-I liver OD in rats	[196]
	Cationic NPs	Antitumorigenic	B16F10 melanoma cells were subcutaneously injected into C57BL/6 mice	[197]
	Nanoencapsulation	Antioxidant, protective against liver and brain damage	As-I liver/brain OD in rats	[198]
	Microcapsules	Anti-inflammatory, antioxidant	AA-I acute colitis in mice	[199]
Silymarin	Liposomes	Hepatoprotective	CCl_4_-I liver damage in rats	[165]
Triptolide	Solid lipid NPs	Anti-inflammatory	Carrageenan-I rat paw edema	[179]
CU	NPs	Hepatoprotective	AcAPh-I hepatotoxicity in rats	[200]
Naringenin	NPs	Hepatoprotective	CCl_4_-I acute rat liver failure	[201]
α-TPh	NE	Anti-diabetes, antioxidant	STZ-I diabetes	[174]
GA	PPC	Anti-apoptotic, cardioprotective	DOX-I cardiac toxicity in rats	[202]
Puerarin	Nano dispersion	Anti-colorectal cancer	HT-29 human colon carcinoma cell subcutaneously injected into BALB/c nude mice	[203]
Resveratrol	Pro-drug	Anti-inflammation	1% DSS-I in drinking water for 8 days mice colon inflammation	[204]
EGCG	Peracetylation	Anti-inflammation, anti-tumorigenesis	DSS-I mice colitis/tumor	[205]

α-TPh = α-tocopherol; GA = gymnemic acid; NPs = nanoparticles; NEs = nano emulsions; EGCG = epigallocatechin gallate; CU = *Cuscuta chinensis*; DENA = diethylnitrosamine; DSS = dextran sulfate sodium; DOX = doxorubicin; AcAPh = acetaminophen; AA = acetic acid; I = induced; As = arsenic; OD = oxidative damage; PPCs = phospholipid complex.

**Table 6 nanomaterials-15-00265-t006:** Main advanced FP types achieved using functional nanomaterials [68].

Advanced FPs Type	Purposes and Description	Key Nanomaterials Currently Used
Physically improved packaging	Packaging materials incorporated with NPs to improve physical properties, such as temperature and moisture stability, mechanical strength, gas barrier, durability, and flexibility	Metal oxides NPs, nanoclays Carbon nanotubes, metallic NPs
Active packaging	NPs endowed with antimicrobial or other functionalities (e.g., antioxidant and UV absorbents) and the ability to release them into packaging. Food packaged into AP results improved in terms of taste, freshness, and shelf life	Ag NPs, Au NPs, metal oxides NPs, antimicrobial and/or antioxidant NPs, functionalized NPs, lignin NPs
Smart packaging	Packaging materials incorporated with nano-sensors to monitor and report on the condition of the food (e.g., oxygen indicators, freshness indicators, and pathogens)	↑ Variability of nano-sensors

↑ = high.

**Table 7 nanomaterials-15-00265-t007:** Main details about the nanosized materials developed in our laboratories in the years 2005–2009.

NPs Type	Size (nm), ζP (mV), PDI EE%, MW *	Bioactivity	Characteristics	APs	Refs.
Sty-CPs **(C8**)	589 nm	Adaptable to biochemical interaction studies with CAOs	Soluble, positive Schiff’s fuchsin-sulfite reagent Spherical morphology	Functionalized glucopyranose	2009 [9]
4-HPR-A-DEX NPs	150–350 nm 0.112–0.176 (PDI)	↑ Cytotoxicity to HTLA-230	CPXs with 4-HPR were prepared by the kneading method TGA suggested. ↑↑ thermal stability, ↑ DL%, ↑ EE% Sustained drug release, possible parenteral administration ↑ Cytotoxicity to HTLA-230 than free 4-HPR ↑ Drug bioavailability, biodegradable	Entrapped 4-HPR	2009 [53]
ATRA-NIC-PVA	<400 nm 0.202–0.450 (PDI)	Cytotoxic to NB cells (LAN-5 cells)	↑ ATRA solubilization and release ↑↑ ATRA aqueous solubility, drug fractional release < 8% ↑↑ Growth inhibition effect than free ATRA Suitable for parenteral injection, drug targeting Long-term storage	Entrapped ATRA	2009 [54]
4-HPR-OL-DEX NPs	~310 nm	In vitro (HTLA-230, LAN-5, and IMR32 NB cells) and in vivo antitumor activity	Tested both in vitro and in vivo, ↑↑ Cytotoxicity in vitro ↑↑ Fraction of sub-G1 cells, no hemolytic activity Suitable for injections, ↑↑ AUC, ↓ clearance ↑↑ Lifespan and long-term survival of treated mice ↑↑ Aqueous solubility, ↑↑ bioavailability	Entrapped 4-HPR	2012 [55]
4-HPR-C_14_-PEG NPs	50–137 nm 0.165–0.221 (PDI)	In vitro (SH-SY5Y and NGP NB cells) antitumor activity	↑↑ Aqueous solubility, suitable for injection Stable aggregates, drug-targeting to solid tumors No release of free 4-HPR in an aqueous environment associated with ↑↑ intracellular concentrations and activity than 4-HPR ↓↓ 4-HPR early metabolism, protracted release	Complexed 4-HPR	2012 [56]
StyGlyco-CCPs **(R1**)	~300 nm	Suitable for interaction with CAOs	Cross-linked resins, spherical morphology	Functionalized D-Glucose	2013 [10]
4-HPR-NGR-NLs	142 nm −19.2 mV 0.073 (PDI) 69% (EE)	In vivo ↑↑ the life of NB mice by apoptotic/anti-angiogenic effects ↓↓ of tumor progression ↓↓ of intra-tumoral vessels ↓↓ of VEGF expression ↓↓ of metalloproteinases MMP2/MMP9	By reverse phase evaporation method ↑↑ Structural integrity of NL in organic fluids Target the tumor endothelial cell marker	Complexed 4-HPR	2013 [57]
StyGlyco-LCPs **(P1G2**)	N.R.	Substrates/inhibitors of CAOs	Soluble	Benlyl amine D-Glactose	2015 [11]
SL[LM-BTZ]	179 nm −33.9 mV 0.070 (PDI)	In vivo ↑↑ the life of NB mice	Lyophilization with cryoprotectants Targeted drug delivery systems, ↑↑ therapeutic index ↑↑ Efficacy, ↑↑ EE%, suitable for intravenous injection ↓↓ of BTZ systemic adverse effects	Complex BTZ	2015 [58]
NGR-SL[LM-BTZ])	173 nm −30.2 mV 0.093 (PDI)				
AN169-PEG-NLs	143.9–153.8 nm 0.052–0.077 (PDI)	In vitro antitumor activity in human cancer cell lines (HTLA-230, Mel 3.0, OVCAR-3, SV620)	By the thin-film hydration method, slow drug release Antitumor activity as free AN169 (72 h) Lyophilization with cryoprotectants Long-term stability, ↑↑ EE%, for intravenous injection	Entrapped AN169	2015 [59]
PolyE-Ds	4.4–5.4 nm 31.2–51.8 mV	Carriers for gene and drug delivery	↑ Water-soluble, excellent β Well-defined sizes, shapes, and ↑ controlled architecture Polycationic, biodegradable, 13,593–25,661 *	Linked amino acids	2017 [12]
*b*-HMPA-Ds.	N.R.	For gene transfection with p-DNA and si-RNA	↑ Water-soluble, excellent β well-defined sizes, shapes, ↑ Controlled architecture, polycationic, biodegradable Not cytotoxic, 9834–60,725*		2017 [13]
PolyE-ADs	3.3–3.6 nm 5.4–5.9 mV	Drug delivery, gene transfection	Well-tailored polymeric structure, excellent β Symmetric tree-like shape, ↑ functional groups Inner cavities, hydrolysable, amphiphilic by a C-18 chain Polycationic, biodegradable, not cytotoxic, 2932–6762 *		2018 [14]
4G PolyE-HDs	4.5–4.6 nm 32.8–33.4 mV	Water-soluble biomedical devices.	Well-tailored polymeric structure; excellent β Symmetric tree-like shape, ↑ functional groups Inner cavities, hydrolysable, hydrophilic; ↑ solubility Polycationic, biodegradable, not cytotoxic, 2932–6762 *		2018 [15]
4G-5G PolyE-DPX	4.4–36.3 nm 15.5–51.8 mV	Several beneficial effects of UOA	Well-tailored polymeric structure; excellent β Symmetric tree-like shape, inner cavities, hydrolysable ↑ Water solubility, polycationic, biodegradable Not cytotoxic, 1360–3130 *	Entrapped UOA	2018 [16]
PolyE-A/H-DPX	18.9–47.7 nm 2.4–15.3 mV	Non-nucleoside HIV-1 reverse transcriptase inhibitor Suitable for parenteral administration	↑ Water-soluble, excellent β, well-defined sizes, shapes, ↑ controlled architecture, polycationic, biodegradable, Not cytotoxic, sustained release, 5851–24,203 *	Entrapped (**1**)	2018 [17]
5G PE-PD-D/GA (GAD)	348.6 nm	Platelet aggregation inhibition ** ROS production inhibition ** Antibacterial (Gram-positive) **	↑ Water-soluble, excellent β, well-defined sizes, spherical morphology, ↑ controlled architecture 192 OH groups, biodegradable, not cytotoxic, 17,010 *	Linked GA	2019 [18]
PolyE-A/H-DPX	60–70 nm	Beneficial effects of EA For clinical applications RSA	300–1000-fold ↑ water solubility, non-PAMAM dendrimers Biodegradable, not cytotoxic, excellent β, 14,287–25,604 * Sustained release, ↑ DL%	Entrapped EA	2019 [19]
5G PE-PD-D/GA (GAD)	348.6 nm	Long-term preservation of EOs	Nano-spherical dendrimer, preservative power ↑ than GA No pro-oxidant action, spherical morphology (SEM) ↑ Compatible with lipids and oily matrices RSA 4-fold ↑ than GA (DPPH), biodegradable by esterase hydrolytic actions, release GA, not cytotoxic, 17,010 *	Linked GA	2019 [20]
5G PE-PD-D/GA (GAD)	348.6 nm	To treat diseases by OS	Nano-spherical dendrimer, no pro-oxidant action Spherical morphology (SEM), RSA 4-fold ↑ than GA, biodegradable (lipase), release GA, not cytotoxic, 17,010 *	Linked GA	2020 [24]
5G-PE-PD-D-OH	44.5 nm −21.2 mV 0.208 (PDI)	ROS-dependent per se cytotoxicity against NB cells sensitive to ETO	Per se activity, nano-spheric, ↑ solubility, 7275 *	Empty dendrimer	2020 [21]
5G PE-PD-DPX (CPX **5**)	70 nm −45 mV	Prevention and treatment of NB cells	↑ Cytotoxicity and pro-oxidant effects than ETO Protection of ETO, synergistic action with ETO Sustained release of ETO, ↑ DL%, ↑ EE Nano-spherical morphology, ↑ solubility	Entrapped ETO	
5G-PE-PD-D-OH	44.5 nm −21.2 mV 0.208 (PDI)	ROS-dependent per se cytotoxicity against NB cells both sensitive and resistant to ETO	↑ Cytotoxicity and pro-oxidant effects than ETO Nano-spherical morphology, 7275 *, 64 peripherals OH Water-soluble	Empty dendrimer	2020 [23]
5G PE-PD-D/GA (GAD)	348.6 nm	Nano-formulation nullifies the pro-oxidant activity of GA To treat diseases by OS To prevent DNA oxidative damage and tumor onset	Nano-spherical dendrimer, no pro-oxidant action Spherical morphology (SEM), RSA 4-fold ↑ than GA Biodegradable (lipase), release GA by hydrolysis Water-soluble, not cytotoxic, 17,010 *	Linked GA	
5G PE-PD-DPX (GALD)	349.9 nm −29.2 mV 0.708 (PDI)		Nano-spherical hygroscopic dendrimer, ↑ water soluble no pro-oxidant action, ↑ DL%, ↑ EE, Spherical morphology (SEM), biodegradable (lipase) Sustained and quantitative release of GA Not cytotoxic, 28,610 *	Entrapped GA	
PolyE-Ds	N.R.	Bactericidal (*Pseudomonas aeruginosa*, *Acinetobacter baumannii*, *Stenotrophomonas maltophilia*)	Non-cytotoxic, amino acid-modified polycationic Ds More potent than colistin against *P. aeruginosa* (5GK) Not lithic behavior, membrane disruptor Broad spectrum of action	Linked K, H, KH	2020 [25]
4-AMSTY-CP (P5)	334 nm +57.6 mV 1.012 (PDI)	Bactericidal (*Enterococcus*, *Staphylococcus*, *Pseudomonas*, *Klebsiella*, *Escherichia coli*, *A. baumannii*, *S. maltophilia*	Random copolymer, ↑ water soluble Rapid (0.5 h) and broad-spectrum non-lytic bactericidal activity, stability in solution, excellent buffer capacity Activity by membrane disruption, 5100 (Mn)	No AP NH_3_^+^ groups	2021 [26]
4-AMSTY-CP (P5)	334 nm +57.6 mV 1.012 (PDI)	ROS-dependent cytotoxic activity on ETO-resistant NB	Random copolymer, ↑ water soluble Stability in solution, excellent buffer capacity Membrane disruptor, cause ↑ ROS generation, 5100 (Mn)	No AP NH_3_^+^ groups	2021 [27]
Sty-CP (P7)	220 nm +49.8 mV 0.809 (PDI)		Random copolymer, ↑ water soluble Stability in solution, excellent buffer capacity Membrane disruptor, cause ↑ ROS generation, 13,719 (Mn)		
4G-5G-PolyE-Ds	16.1–24.9 nm +24.8–34.0 mV	↑ Antibacterial effects (MIC = 0.5–8.7 µM vs. Enterococci and Staphylococci)	Activity depended on the density and on the type of cationic amino acid-conjugated dendrimers and not on the presence and the release of UOA, ↑ water soluble Stability in solution, excellent buffer capacity Protracted release of UOA, membrane disruptor 14,600–29,300 *	Linked K, R, KR Entrapped UOA	2021 [28]
4G-BBB4-PolyE-Ds	112.1 nm +28.9 mV 0.289 (PDI)	Antibacterial vs. Staphylococci To treat skin infections	↑↑↑ Water solubility than BBB4, good DL and EE SI = 1.4–5.5, protracted release of BBB4 Membrane disruptor ↓ Cytotoxicity on HaCat than BBB4, 21,176 *	Linked K Entrapped BBB4	2021 [29]
		N.D.	↑↑↑ Water solubility than BBB4, good DL and EE Sustained/protracted release of BBB4, 21,176 *	Linked K Entrapped BBB4	2021 [30]
RES-TPGS	9.6–12.7 nm ^§^ −1.6—4.8 mV ^§^ 0.13–0.26 (PDI) ^§^	Antioxidant Anti-inflammatory Protective action in the liver	Micellar NPs, ↑↑↑ water solubility than RES Good DL% and EE% Sustained/protracted release of RES ↓ Cytotoxicity on HaCat than RES, 21176	Entrapped RES	2021 [31]
UA-4G PolyE-D (UA-G4K NPs)	577.5 vs. 333.4 nm 4GK −42.6 vs. +66.1 mV 4GK 0.235 vs. 0.286 4GK (PDI)	N.D.	Biodegradable, not cytotoxic (HeLa cells), ↑ DL%, spheric Protracted release profile governed by diffusion Water solubility 1868-fold ↑ than UA Clinical applicability, 30,069 *	Linked K Entrapped UA	2021 [34]
		Antibacterial vs. enterococci (MICs = 0.5–4.3 µM) Bactericidal vs. *E. faecium*	↓ Cytotoxicity on HaCat (IC_50_ 96.4 µM), SIs = 22–193 Valuable as a novel oral-administrable therapeutic option to treat enterococcal infections.		2021 [32]
ATRA-TPGS NPs	14.1–21.0 nm ° −7.2–−13.0 mV ° 0.19–0.32 (PDI, water) °	To prepare topical gel Treatment for skin diseases Treatment for melanoma	↓ATRA cutaneous side effects, ↑ stability than ATRA. ↑ ATRA solubilization, good EE% 22 ± 4 µ cm^−2^ permeation after 24 h ↑ Cytotoxic effects on melanoma cells	Entrapped ATRA	2021 [33]
Sty-CP (P7)	220 nm +49.8 mV 0.809 (PDI)	Antibacterial activity vs. enterococci, staphylococci *Acinetobacters*, *Pseudomonas Klebsielle*, *Escherichia coli* *Stenotrophomonas maltophylia* Rapid bactericidal effects on *S. aureus, K. pneumoniae,* and *P. aeruginosa*	Random copolymer, ↑ water soluble ↑ Stability in solution, excellent buffer capacity Membrane disruptor, ↓tendency to develop resistance ↓toxicity, long-term activity, 13,719 (Mn) Lowest MICs = 0.6–1.2 µM	No AP NH_3_^+^ groups	2021 [35]
5G-PolySty-D (5G-PDK)	203.0 nm +19.2 mV 0.282 (PDI)	Rapid bactericidal effects vs. *A. baumannii*, *A. pittii* *A. ursingii*	Membrane disrupters, MICs = 3.2–12.7 µM Electrostatic interactions with bacterial surfaces Self-biodegradable, 20,145.3 *	64 Linked K	2021 [36]
		Antibacterial and bactericidal vs. Pseudomonadaceae	MICs depending on pigment production *P. aeruginosa* = 1.6-> 6.4 µM *P. putida* producing pyoverdine = 3.2–6.4 µM *P. putida* producing non-pigmented colonies = 0.2–1.6 µM ↓Cytotoxicity on HaCat, ↑↑↑ SIs (13–404)		2021 [37]
4-AMSTY CP (CP1)	833.4 nm +27.3 mV 0.2235 (PDI) 157,306 *	Potent broad-spectrum antibacterial effects Kill pathogens rapidly	Cationic macromolecules acting as membrane disruptors: **CP1** MICs = 0.1–0.8 µM, **OP2** MICs = 0.35–2.8 µM Promising ingredients for the development of novel antibacterial dosage forms for topical applications (hydrogel) Spherical morphology	No AP NH_3_^+^ groups	2022 [38]
4-AESTY OP (OP2)	163.4 nm +31.1 mV 0.301 (PDI) 44,514 *				
CB1H-P7 NPs	142.9 nm +36.7 mV 0.626 (PDI)	N.D.	Spherical morphology, positive surface charge ↑↑↑ DL%, ↑↑↑ EE%, protracted release profile, 26,623.9 *	CB1H	2022 [39]
		Antibacterial vs. G+/G- Bactericidal vs. *S. aureus* *E. coli*, *P. aeruginosa*.	MICs ↓↓ of pristine CB1H and matrix P7 NPs displayed MICs = 0.6–4.8 µM on 34 out of 36 isolates. ↓Cytotoxicity on HaCat, SIs up to 2.4		2022 [43]
		New curative option vs. NB (IMR 32 and SHSY 5Y cells)	Membrane disruptors, IC_50_ = 0.43–0.54 µM vs. IMR 32 and SHSY 5Y cells Early-stage (66–85%) and late-stage apoptosis (52–65%) Effects of CB1H and P7 ↑ by 54–57 and 2.5–4 times (IMR32) ↑ By 53–61 and 1.3–2 times against SHSY 5Y 1–12-fold more potent than fenretinide ↓Cytotoxicity on HaCat, SIs = 2.8–3.3		2023 [46]
CR232-SUVs	173.4 nm +17.8 mV 0.118 (PDI)	N.D.	Biocompatible, DL%, EE% ↑ with ↑ lipids/CR232 ratio Prolonged release profile ruled by zero-order kinetics 1764-fold more soluble than the untreated CR232	Linked K Entrapped CR232	2022 [40]
5GK PoliE-D NPs (CR232-G5K NPs)	529.7 nm +37.2 mV 0.472 (PDI)	N.D.	2311-fold more water-soluble than pristine CR232 No use of harmful organic solvents/additives, spheric Biodegradable, ↑↑↑ DL%, ↑↑↑ EE%, ↓Cytotoxicity on HeLa Quantitative release profile (Weibull kinetics), 44,153.1 *		
		Antibacterial vs. G+ and G-Rapid bactericidal activity	MICs = 0.36–2.89 µM vs. all of the considered G+ and G- MICs = 0.72 µM vs. colistin-resistant *P. aerginosa* and *K. pneumoniae* carbapenemases (KPCs)-producing ↓Cytotoxicity on HaCat, SIs up to 8		2022 [41]
4-HPR-P5	249 nm +41.3 mV 0.210 (PDI)	Antiproliferative activity IC_50_ = 1.25 µM (IMR32), 1.93 µM (SH-SY5Y)	Molecularly dispersed 4-HPR using P5 as a solubilizing agent by the antisolvent co-precipitation method ↑↑↑ Clinical outcomes of 4-HPR 4-HPR apparent solubility 1134-fold ↑, faster dissolution suitable for intravenous administration, ↑↑↑ DL% Extended release over time, excellent β	Dispersed 4-HPR	2023 [45]
P5PA-4I NPs	541 nm +8.39 mV 0.194 (PDI)	New promising treatment for chemo-resistant NB Cytotoxic to ETO-sensitive (HTLA-230) and to ETO-resistant (HTLA-ER) cells	↑↑↑ Activity of 4I, ↑↑↑ DL%, ↑↑↑ EE% ↑↑↑ hydrophilic–lipophilic balance (HLB) Excellent buffer capacity, ↑↑↑ residence time inside cells. Chemically stable in an aqueous medium > 40 days Assumed low hemolytic toxicity ROS-dependent cytotoxic effects ↑↑↑ than 4I ↑↑↑ Efficacy than ETO in HTLA-ER cells	Loaded 4I	2023 [47]
TPP-BA-NVs	49.3 nm +18.2 mV 0.529 (PDI) 852.7 *	Cytotoxic to MDR HR-NB IC_50_ = 0.2 µM (HTLA-230) IC_50_ = 1.1 µM (HTLA-ER)	Tested on HTLA-230 human stage-IV NB cells and HTLA-ER NB cells resistant to ETO, DOX, etc. IC_50_ = 538-fold ↓ than ETO (HTLA-ER) Limited cytotoxic effects against mammalian cell lines. (Cos-7, IC_50_ = 4.9 µM, HepG2, IC_50_ = 9.6 µM, MRC-5, IC_50_ = 2.8 µM, RBCs, IC_50_ = 14.9 µM, SIs = 2.5–74.6	No additional AP Linked BPPB	2024 [49]
		↑↑↑ Antibacterial effects on 50 G+ and G− MDR and ESKAPE pathogens	Characterization of BPPB by ATR-FTIR, NMR, UV, FIA-MS (ESI), elemental analysis, and potentiometric titrations. Spherical vesicles, MICs = 0.250–32 µg/mL, SIs > 10		2024 [50]
		New treatments for CMM by MeOV and MeTRAV IC_50_ = 49 nM on MeOV (72 h)	Cytoplasmic membrane disruptors triggering OS ROS-correlated apoptotic effects ↓ Cytotoxicity to non-tumoral cells and RBCs SIs up to 299 on MeOV (72 h)		2024 [52]
			Biodegradable HA-based hydrogel formulation HA-BPPB-HA possesses ↑↑↑ swelling capability ↑↑↑ Porosity, viscous elastic rheological behavior		
CP5 (P5)/DMAA 11b,c/DMAA CP MA/DMAA CP	334, 2590, 373, 112 nm +58, +6.5, +25, +18 mV 1.012, 0.281, 0.326, 0.590^PDI^	Possibility to develop M21 as a new scaffold for TE	Amine- or aldehyde-containing CPs were developed CPs by 5 (P5), 11b, and 11c are excellent substrates for LO CPs by 5, 11b, and 11c and MA cross-linked Gel B M21 by P5/DMMA has 71% cross-linking M21 is biocompatible	NH_3_^+^ groups CHO groups	2024 [51,216]
4-HPR-TPGS-DSPE-PEG	11.4–15.7 nm −4–−14 mV 0.12–0.46 (PDI)	Cytotoxic to MDR HR-NB	Micelles prepared using the solvent casting technique Good DL%, ↑↑↑ EE %, stable colloidal dispersions Apparent solubility 363-fold ↑ than 4-HPR Slow-release behavior of about 28% (24 h) ↑ Cytotoxicity than 4-HPR on SK-N-BE-2C NB cells	Entrapped 4-HPR	2024 [217]

APs = Active principles; CAO = copper-containing amine oxidases; Sty-CPs = Styrene-based copolymers; StyGlyco-CCPs = Glyco-styrene-based cross-linked copolymers; StyGlyco-LCPs = Glyco-styrene-based linear copolymers; N.R. = not reported; PolyE-Ds = polyester-based dendrimers; ↑ = high, highly; ζP = Zeta potential; PDI = polydispersity index; * = molecular weight; b-HMPA-Ds.= polyester-based dendrimers based on 2, 2-bis-(hydroxymethyl)-propanoic acid; PolyE-ADs = poly-ester-based amphiphilic dendrimers; 4G PolyE-HDs = fourth-generation polyester-based hydrophilic dendrimers; β = buffer capacity; UOA = ursolic mixed with oleanolic triterpenoid acids; 4G-5G PolyE-DPX = G4-G5 polyester-based dendrimer dendriplexes loaded with UOA; PolyE-A/H-DPX = polyester-based hydrophilic and amphiphilic dendriplexes (DPX) loaded with insoluble (**1**) (thiocarbamate derivative); GA = gallic acid; 5G PE-PD-D/GA (GAD) = 5G polyester-based propane diol (PD) dendrimer linked to GA; ** 7–50-fold more active than GA alone; EA = ellagic acid; PolyE-A-DPX = polyester-based amphiphilic/hydrophilic DPX loaded with the insoluble EA; DL = drug loading; RSA = radical scavenging activity; EOs = essential oils; OS = oxidative stress; 5G-PE-PD-D-OH = polyester-based propane diol dendrimer with 64 peripheral OH; ETO = etoposide; 5G PE-PD-DPX (CPX **5**) = polyester-based propane diol dendriplex loaded with ETO; CPX = complex; NB = neuroblastoma; 5G PE-PD-DPX (GALD) = polyester-based propane diol dendriplex loaded with GA; EE = encapsulation efficiency; K = lysine; H = histidine; 4-AMSTY-CP = 4-ammoniumbuthylstyrene-based random copolymer; R = arginine; BBB4 = 2-(4-bromo-3,5-diphenyl-pyrazol-1-yl)-ethanol; SI = selectivity index; N.D. = not yet determined; RES-TPGS = D-α-tocopheryl-polyethylene-glycol-succinate micelles loaded with resveratrol (RES); HaCaT = human keratinocytes; § = fresh loaded; UA-4G PolyE-D = fourth-generation lysine polyester dendrimer (4GK9) loaded with ursolic acid (UA); ATRA = all-*trans*-retinoic acid; ATRA-TPGS NPs = D-α-tocopheryl-polyethylene-glycol-succinate (TPGS) NPs loaded with ATRA; ° = lyophilized; MICs = minimal inhibitory concentrations; 5G-PDK = fifth-generation lysine-modified cationic polyester-based dendrimer with a propane diol (PD) core; 4-AMSTY CP = 4-ammoniummethylstyrene copolymer; 4-AESTY OP = 4-ammoniumethylstyrene homopolymer; CB1H = pyrazole derivative as hydrochloride salt; CB1H-P7 NPs = copolymer P7 loaded with CB1H; G+ = Gram-positive; G- = Gram-negative; CR232 = 3-(4-chlorophenyl)-5-(4-nitrophenylamino)-1*H*-pyrazole-4-carbonitrile; 5GK PoliE-D NPs = fifth-generation polyester-based dendrimer containing lysine (5GK) loaded with CR232; CR232-SUVs = CR232-loaded liposomes; 4-HPR = fenretinide; 4-HPR-P5 = solid dispersion of 4-HPR using P5; 4I = synthesized imidazo-pyrazole (IMP); PA = palmitic acid; 4I-loaded cationic NPs achieved by using P5 and PA as nanosized matrices; TPPs = triphenyl phosphonium salts; TPP-BA-NVs = TPP-based bola amphiphilic (BA) nanovesicles (NVs); HR-NB = high-risk NB; BPPB = sterically hindered quaternary *bis*-phosphonium bromide; DOX = doxorubicin; CMM = cutaneous metastatic melanoma; HA = hyaluronic acid; 11b, c = acryloyl amidoamine monomers; MA =methacrolein monomer; DMAA = dimethylacrylamide; CP5 (P5)/DMAA = ammonium CP of P5 with DMAA; 11a, c/DMAA CPs = ammonium acrylic CPs of 11b and 11c with DMAA; MA/DMAA CP = aldehyde CP of MA with DMAA; LO = lysil oxidase; TE = tissue engineering; DSPE-PEG = 1,2-dis-tearoyl-glycero-3-phosphoethanola-mine-N-[methoxy(polyethyleneglycol)-2000]; 4-HPR-TPGS-DSPE-PEG = mixed micelles made of TPGS and DSPE-PEG loaded with 4-HPR; PCLX = paclitaxel; CPTC = camptothecin; 4-HPR-A-DEX NPs = amphiphilic dextrins NPs loaded with 4-HPR; TGA = thermogravimetric analysis; NIC = nicotinoyl; ATRA-NIC-PVA = amphiphilic polymeric micelles made of NIC-esterified polyvinyl alcohol (PVA) complexed with ATRA; 4-HPR-OL-DEX NPs = amphiphilic dextrin oleate NPs loaded with 4-HPR; PEG = polyethylene glycol; *b*PEG = branched PEG; 4-HPR-C_14_-*b*PEG NPs = alkylated *b*PEG micelles loaded with 4-HPR; 4-HPR-NGR-NLs = nanoliposomes (NLs) functionalized with NGR peptides and loaded with 4-HPR; SL[LM-BTZ] = stealth liposomes (SLs) complexed with an amino-lactose (LM) and loaded with bortezomib (BTZ); NGR-SL[LM-BTZ] = stealth liposomes (SLs) functionalized with NGR peptides, complexed with an amino-lactose (LM) and loaded with bortezomib (BTZ); AN169-PEG-NLs = pegylated nanoliposomes loaded with a naphthalenediimide derivative (AN169); ↓ = reduced, decreased, low, lower; ↑ improved, increased, high, higher. More than one arrow means a higher effect.

**Table 8 nanomaterials-15-00265-t008:** Some general findings achieved by evaluations made on some NPs.

Food/Simulant	NPs/Nanocomposite	Additive/Fortifier	Migration	Ref.
Vegetables	Starch/clay	None	In conformity with European directive	[68]
Fatty food simulant *	PLA/laurate	LHD-C12	Below legal migration limits	[223]
Food simulant	CNT/LDPE/PS	None	No migration	[224]
Chicken meatballs	AgNPs **	None	Slow migration	[225]
Chicken breast	AgNPs/co-PEFs	None	No transfer	[226]
Distilled water				
Acidic food simulants ***	AgNPs/PE	Irganox 1076 Irgafos 168 Chimassorb 944 Tinuvin 622 UV-531, UV-P	Transfer promoted by organic additives	[227]

LDH-C12 = layered double hydroxide (LDH-C12); * to predict possible migration in meat; CNT = carbon nano-tube NPs; LDPE = low-density polyethylene; PS = polystyrene; ** present in an already marketed FP; PEF = polyester films; *** 3% acetic acid; PE = polyethylene.

**Table 9 nanomaterials-15-00265-t009:** Detailed migration results expressed as wt/volume or as wt/wt of numerous inorganic NPs from different food contact materials (FCMs) into food or food simulants.

NPs	Polymer	FCMs	Migration Result
ZnONPs	LDPE	Films	0.009–3.416 mg/L
TiO_2_ NPs	PET	Films	1.88–3.32 ng/kg
AgNPs	LDPE	Baby products	1.05–2.25 ng/L
Nanoclay	LDPE-EVA	Films	N.D.
AgNPs	LDPE	Commercial cutting board	0.24–0.60 µg/g
TiO_2_ NPs	PLA	Films	2.19–3.5 µg/kg
TiO_2_ NPs/AgNPs	PLA	Films	2.36 µg/kg
TiO_2_ NPs/AgNPs		Films	0.593–0.8 µg/kg
graphene	LDPE	Films	1.02–1.29 MG/kg
TiO_2_ NPs	LDPE	Films	0.61 mg/kg
ZnONPs	LDPE	Films	14.17 mg/kg
ZnONPs	LDPE	Plaques	0.05–2 mg/kg
AgNPs	PP	Two plastic containers	62–18,887 ng/dm^2^
	PC	Baby feeding bottle	
		Food pox	
AgNPs	PE, HDPE	Food storage boxes Commercial storage boxes	<0.04–0.31 µg/g
		Commercial container Commercial bags	0.5–46 µg/L
ZnONPs	PE, HDPE	Commercial containers, bags, dishes, cups	0.54–46 µg/L
AgNPs	PE	Commercial containers	3.17–5.66 µg/L
	PE with a 10 µm AgNPs coating	Commercial cling films	0.01–28.92 µg/L
	PP, LDPE, PS	Baby bottles Cutting boards Food storage bags Food storage containers	6.60–35.8 μg/g
AgNPs	PP, LDPE	Commercial food containers	<0.0001–0.1 ng/g
SiNPs	LDPE	Films	N.D.
Cloisit 20 A	PET	Bottle	0.18–9.5 mg/kg
Carbon black	LDPE, PS	Injection-molded plaques	N.D.
TiN	LDPE	Films	0.09–0.24 μg/kg
AgNPs	PE	Films	0.003–0.005 mg/dm^2^
	LDPE		0.30–1.43 mg/kg
CuNPs	PE	Films	0.024–0.049 mg/dm^2^
TiO_2_ NPs	PE	Films	0.5–12.1 µg/kg
AgNPs	PP, PE	Commercial plastic container	4.75–9.5 ng/cm^2^
AgNPs	Plasticized PVC	Commercial plastic bags	0.5 ng/cm^2^
		Film	0.01–0.37 mg/dm^2^
AgNPs	LDPE	Commercial bags, containers	3.1 × 10^−3^–3.74 ng/cm^2^
	PP	Commercial bags, containers	50.3 × 10^−3^–31.46 ng/cm^2^
	PE	Commercial bags	1–4 µg/dm^−2^
AgNPs	PE	Commercial food contact film	0.22–5.6%
ZnONPs	LDPE	Film	0.11–0.68 μg/L
Cloisite	PLA	Film	N.D.

N.D. = not detected.

**Table 10 nanomaterials-15-00265-t010:** Results from in vitro studies on possible toxic effects on different cell lines of different NPs at different concentrations having different morphologies (nano wires, nano roads, nano spheres, etc.).

NPs	Size/Shape/Concentration	Effects on Cells	Refs.
Ag NPs	2–8 nm	Genotoxic and cytotoxic effects on root meristematic cells of *Allium cepa* (A. cepa) ↓ Mitotic index ↑ Chromosomal aberration number	[231]
Ag NWs	20,000 × 65 nm 0.39–25 μg/mL	Less toxic than nanoplates Toxicity was not only caused by Ag^+^ release No established LC_50_ value	[232]
	100 nm 4 μg/cm^2^	Toxicity to human monocyte-derived macrophage THP-1 cells ↓ Cell proliferation, ↑ increase in membrane instability	
	40 nm 5–30 μg/mL	Toxicity to RBCs Cell deformability, aggregation, and hemolysis in a dose-dependent manner	
Ag NSs	30 nm 0.05–5 μg/cm^2^	Cytotoxic and genotoxic to fish OLHNI2 cells Chromosomal aberrations	
	10, 20, 40 nm 0.39–25 μg/mL	Cytotoxicity and superoxide generation in a fish gill cell line ↓ Toxic than nanoplates None of these contributions established an AgNW LD or LC_50_ value	
SPM Fe NPs	> 100 µg/mL	Impaired DNA, nucleus, and mitochondria in different cell lines Causes ↑ ROS, inflammation	[233]
Fe NWs	50 nm 10,000 NWs per cell	Toxicity to HeLa cells No significant effect. MTT assay Up to 10,000 NWs per cell (72 h) ↑ cell viability of about 80%.	[232]
Zr NPs	5–30 nm	↑ Viral receptor expressions inflammation	[234]
Ce NRs	>200 nm	Progressive pro-inflammatory effect and cytotoxicity in THP-1 cells	[232]
HAP	Crystals, H-rod H-needle, H-sphere H-plate	Decreased cell viability and consequent necrosis in rat aortic smooth muscle cells	[235]
AuNPs	5 nm	Toxicity, ↑ cytokine production in mouse fibroblasts	[236]
Au NRs	54 nm 30–100 μM	Toxicity to human prostate cancer cell line DU145, cervix carcinoma cell line HeLa, and male C57/BL6 mice No genotoxicity, induction of autophagy Destabilization of lysosomes, alterations of actin cytoskeleton Impairments in cell migration	[232]
	65 nm 0.5 mM	No toxicity in HeLa cells > 90% viability after 24 h	
Ni NWs	33 nm diameter 5 μg/mL	↓ Viability in human colorectal carcinoma HCT 116 cells	[237]
	200 nm 106 NPs/mL	Toxicity to rat marrow stromal cells (MSCs), MC3T3-E1 osteoblast cells, and UMR-106 osteosarcoma cells Binding to cytoplasmic metalloproteins Trigger lysosome reorganization around the nucleus Cell viability was more than 95% up to 5 days after internalization	[232]
	200 nm 35,000 NPs/mL	Cytotoxicity to L929 mouse fibroblast cells No cytotoxicity.	
	33 nm 5 μg/mL	Cytotoxicity to HCT 116 cells Viability of HCT 116 cells ↓↓ at 24, 48, and 72 h exposure	
Al_2_O_3_ NWs	200–400 nm 50–200 mg/mL	Viability of L929 and RAW264 cells was not ↓↓, no ↑ LDH release Not cytotoxic, no nuclei damage	[232]
Zn NPs	4–20 nm	Low viability, ROS production, and cytotoxicity in human immune cells	[238]
Ti NPs	70 nm 50 μg/mL	Inflammation ↑ IL-8 in human microvascular endothelial cells	[239]
	3/600 μg/mL	Shrinking of cells, lower metabolic activity, releasing of LDH ROS production in mouse fibroblast L929	[240]
	70 nm	↑ Viral receptor expressions and inflammation	[234]
Ti NWs	10 μg/mL	RAW264.7, H9C2, Chang human liver cells HACAT, MH-S, HEK-293, TM3, BEAS-2B cells Toxicity depended on the surface area of TiO_2_ NWs	[232]
	<10 nm 12.5–350 μg/mL	Toxicity to Caco-2/HT29 intestinal cells Non-cytotoxic damage was detected (24 h) Viability was above 80% Different interactions and cellular responses related to differently shaped TiO_2_ NPs	
Ti NRs:	<100 nm 12.5–350 μg/mL		
Ti NSs	25 nm 12.5–350 μg/mL		
GO NPs	Up to 25 μg/mL	Effected antigen inhibition ↓ Intracellular levels of immune proteasome	[241]
Co NPs	50–200 nm	Pro-inflammatory effect on naïve macrophages ↓ Anti-inflammatory IL-1Ra ↑ Inflammatory TNF-a	[234]
Co NWs	12.5–175 μg/mL	Apparent cytotoxicity to 3T3 and 4T1 cells after 9 h (50 μg/mL)	[232]
MSP Si NPs	100 nm	Membrane deformities and hemolysis in RBC	[242]
	15 nm	Strongly biased naïve macrophages towards inflammation ↑ Inflammatory cytokines IL-1b, TNF-a. ↑ Inflammatory phenotype of LPS-polarized M1 macrophages	[234]
Si NWs	100 nm 6.25–100 μg/mL	Pre-osteoblast subclones (MC3T3-E1) cells Induce apoptosis due to OS in MC3T3-E1 cells (48 h) Cell viability remains ↑ after 24 h	[232]
	50/150 mg/mL	HeLa, HepG2, HEK293T, and human normal liver-7702 cells Cytotoxicity ↑ dependent on cell lines, concentration, and incubation time	
Cloisite^®^ Na^+^	0–125 mg/mL	Cytotoxicity and mutagenicity in the HUVEC No cytotoxic or mutagenic effect	[229]
Cloisite^®^130B	0–250 mg/mL	Cytotoxicity and mutagenicity in the HUVEC toxic effects	

NRs = nano roads; NW = nano wires; SPM = superparamagnetic; HAP = hydroxyapatite; GO = graphene oxide; MSP = mesoporous; OS = oxidative stress; NSs = nano spheres; RBC = red blood cells; ↓ = reduced, decreased, low, lower; ↑ improved, increased, high, higher; HUVEC = human umbilical vein endothelial cells.

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
