# Peer review of "Last Fifteen Years of Nanotechnology Application with Our Contribute"

_nanomaterials, 2025, doi:10.3390/nano15040265_

Round 1

Reviewer 1 Report

Comments and Suggestions for Authors

  After reading this manuscript, I think the topic is not specific and not objective. It is not suitable to give the work summary of one group as review. It is better to be focused on some issues to review relevant reference. This manuscript might be considered as comment or report, not at its present form.

Comments on the Quality of English Language

English should be improved.

Author Response

 After reading this manuscript, I think the topic is not specific and not objective. It is not suitable to give the work summary of one group as review. It is better to be focused on some issues to review relevant reference. This manuscript might be considered as comment or report, not at its present form.

We apologize in advance to the Reviewer for our disagreement with the Reviewer comment. Differently from what asserted by the Reviewer, our manuscript does not concern only a summary of the work of one group (in this case our group). This content constitutes only a limited part (less than 8 pages out of 73) of the manuscript, specifically Section 3 containing Table 7 (revised manuscript), as a rapid summary of our research in Nanotechnology application during the last 15 years. The rest of manuscript (more than 65 pages out of 71) explores and discusses several aspects of nanotechnology applications. It contains an introductive part encompassing surveys made using Scopus database to investigate the number of works on nanomaterials of different types and different applications published in the last fifteen years, as an index of the expansion of nanotechnology in the last years. The statistical analysis of data has been reported into two graphs. A discussion on the current economic impact and industrial involvement of nanomaterials and on that observed along the years, with a new graph, have now included into the Section 1 of the manuscript. Following, the manuscript contains Section 2, made of 7 subsections, Figure 4, 5, as well as 6 Tables (revised version). Here the main types of nanomaterials and their applications to bioactive natural and synthetic principles, food constituents, phytochemicals, nutraceutical or in the food packaging sector, and the main technique used to characterize NPs have been reported and discussed. Then, after Section 3, an extensive discussion has been devoted to the still unsolved question of nanotoxicology (Section 4), containing one subsection with 4 Tables and Figure 6, followed by Section 5 as conclusions.

Reviewer 2 Report

Comments and Suggestions for Authors

I am not able to review the more technical part on nanomaterials, so my comments are exclusively about the presentation of results.

The authors should insert a "Materials and methods" part in which you should clearly present how you chose the paper to include in the review, inclusion and exclusion criteria, what did you analyzed...

You could use PRISMA guidelines https://www.prisma-statement.org/ to better structure the work. It is a standard format and it si preferrable in this type of papers.

Author Response

I am not able to review the more technical part on nanomaterials, so my comments are exclusively about the presentation of results.

The authors should insert a "Materials and methods" part in which you should clearly present how you chose the paper to include in the review, inclusion and exclusion criteria, what did you analyzed...

You could use PRISMA guidelines https://www.prisma-statement.org/ to better structure the work. It is a standard format and it si preferrable in this type of papers.

We thank the Reviewer for his/her comment that offered us the possibility to better clarify how we

chose the papers to for editing this review, inclusion and exclusion criteria, etc. The Reviewer’s

request has been addressed introducing a new 1.3 subsection in Section 1. Please, see lines 182-200.

Reviewer 3 Report

Comments and Suggestions for Authors

It is a huge paper (in my oppinion to huge) reviewing the achievements of Authors on the field of nanotechnology, mostly nanobiotechnology. This topicisis discussed against the broader background and shown significant contribution of the Authors here. Thus, paper could be published as it i in this special volumes

Author Response

It is a huge paper (in my oppinion to huge) reviewing the achievements of Authors on the field of nanotechnology, mostly nanobiotechnology. This topicisis discussed against the broader background and shown significant contribution of the Authors here. Thus, paper could be published as it i in this special volumes.

We thank a lot the Reviewer for his/her positive comments and appreciations.

Reviewer 4 Report

Comments and Suggestions for Authors

The manuscript titled “Last Fifteen Years of Nanotechnology Application with Our Contribute” by Alfei, S.; et al. is a Review work where the authors outlined the most recent advances in the field of Nanotechnology and the use of Nanomaterials in the design of the next-generation of strategies to fight against human diseases and the production of green-friendly food packaging materials. This Review is interesting but is required to enlarge the list of potential smart applications found in their explotaitation to strengthen the relevance of this field on society. The manuscript is generally well-written and this is a topic of growing interest.

However, it exists some points that need to be addressed (please, see them below detailed point-by-point) to improve the scientific quality of the submitted manuscript paper before this article will be consider for its publication in Nanomaterials.

1) The authors should consider to add the term “nanoscience revolution” in the keyword list.

2) “Currently, nanotechnology is the most promising science, engineering, and technology conducted at the nanoscale and is used in several sectors” (lines 35-36) and “2. Among the Nanotechnology Applications” (lines 104-588). Could the authors provide quantitative data insights according to the worldwide economic impact associated to the nanomaterials manufacturing by the different Industrial sectors? This will significantly aid the potential readers to better understand the significance of this devoted Review work.

3) Then, in the previously described section titled “2. Among the Nanotechnology Applications” (lines 104-588) the authors are only focused in the application development in the sector of drug delivery and nanomaterials formulation. It is necessary to expand the discussion to other alternative applications in the use of nanomaterials. In this framework, the development of quantum technologies [1] or the production of smart coating surfaces with enhanced antimicrobial properties [2] need to be mentioned.

[1] Winkler, R.; et al. A Review of the Current State of Magnetic Force Microscopy to Unravel the Magnetic Properties of Nanomaterials Applied in Biological Systems and Future Directions for Quantum Technologies. Nanomaterials 2023, 13, 2585. https://doi.org/10.3390/nano13182585

[2] Zhang, T-G.; et al. Iron Oxide Nanoparticles as Promising Antibacterial Agents of New Generation. Nanomaterials 2024, 14, 1311. https://doi.org/10.3390/nano14151311

4) “2.1.2. Main NPs Developed to Nano Formulate Natural and Synthetic Aps” (lines 178-588). A schematic representation will benefit to the potential readers to better visualize the existing nanoformulation routes recently reported in literature.

5) Then, the main characterization techniques to ascertain the nanoparticle and nanomaterial properties with their inherent advantages and limitations need to be extensively discussed. This will support the potential readers in the decision-making process and the experimental flowchart to follow in this field.

6) “More in Deep about Nanotechnology Applications in Food-Packaging (FP) Industry” (lines 554-588). First, this subsections requires to be numbered. The authors need to check the rest of the subsections because this issue was repeated not only here. Then, the content of this subsection should be enlarge by the potential use of lignin nanoparticles to evolve suistainable food packaging materials.

7) “5. Conclusions” (lines 822-864). This section perfectly remarks the most relevant outcomes found by the authors in this field and the promising future prospectives. It may be desirable to add a brief statement to also highlight the future action lines to pursue the topic covered in this topic.

Author Response

The manuscript titled “Last Fifteen Years of Nanotechnology Application with Our Contribute” by Alfei, S.; et al. is a Review work where the authors outlined the most recent advances in the field of Nanotechnology and the use of Nanomaterials in the design of the next-generation of strategies to fight against human diseases and the production of green-friendly food packaging materials. This Review is interesting but is required to enlarge the list of potential smart applications found in their explotaitation to strengthen the relevance of this field on society. The manuscript is generally well-written and this is a topic of growing interest.

However, it exists some points that need to be addressed (please, see them below detailed point-by-point) to improve the scientific quality of the submitted manuscript paper before this article will be consider for its publication in Nanomaterials.

1) The authors should consider to add the term “nanoscience revolution” in the keyword list.

We thank the Reviewer for this suggestion, which has been addressed in line 35.

2) “Currently, nanotechnology is the most promising science, engineering, and technology conducted at the nanoscale and is used in several sectors” (lines 35-36) and “2. Among the Nanotechnology Applications” (lines 104-588). Could the authors provide quantitative data insights according to the worldwide economic impact associated to the nanomaterials manufacturing by the different Industrial sectors? This will significantly aid the potential readers to better understand the significance of this devoted Review work.

We thank the Reviewer for this suggestion. His/her suggestions have been addressed by adding new paragraphs in Section 1, with 7 additional references (lines 38-85), creating a new Section 1.1. (lines 86-110) including a new graph (Figure 1) and a new reference. The abstract was slightly modified accordingly.

3) Then, in the previously described section titled “2. Among the Nanotechnology Applications” (lines 104-588) the authors are only focused in the application development in the sector of drug delivery and nanomaterials formulation. It is necessary to expand the discussion to other alternative applications in the use of nanomaterials. In this framework, the development of quantum technologies [1] or the production of smart coating surfaces with enhanced antimicrobial properties [2] need to be mentioned.

[1] Winkler, R.; et al. A Review of the Current State of Magnetic Force Microscopy to Unravel the Magnetic Properties of Nanomaterials Applied in Biological Systems and Future Directions for Quantum Technologies. Nanomaterials 2023, 13, 2585. https://doi.org/10.3390/nano13182585

[2] Zhang, T-G.; et al. Iron Oxide Nanoparticles as Promising Antibacterial Agents of New Generation. Nanomaterials 2024, 14, 1311. https://doi.org/10.3390/nano14151311

The alternative applications suggested by the Reviewer have been included in Section 2.1. with related references. Please, see lines 207-233.

4) “2.1.2. Main NPs Developed to Nano Formulate Natural and Synthetic Aps” (lines 178-588). A schematic representation will benefit to the potential readers to better visualize the existing nanoformulation routes recently reported in literature.

We thank the Reviewer for the suggestion, which has been addressed by inserting a new Figure 4, and a few lines of explanation. Please see lines 303-313. However, we make kindly note to the Reviewer that a series of Figures (Figure 1A-6A) were already present in the original form of the manuscript in Appendix A at the end of the manuscript, which probably could satisfy his/her request.

5) Then, the main characterization techniques to ascertain the nanoparticle and nanomaterial properties with their inherent advantages and limitations need to be extensively discussed. This will support the potential readers in the decision-making process and the experimental flowchart to follow in this field.

The useful suggestion of the Reviewer has been addressed by adding the new Section 2.1.3, two new Tables and related references. Please see lines 628-647.

6) “More in Deep about Nanotechnology Applications in Food-Packaging (FP) Industry” (lines 554-588). First, this subsections requires to be numbered. The authors need to check the rest of the subsections because this issue was repeated not only here. Then, the content of this subsection should be enlarge by the potential use of lignin nanoparticles to evolve suistainable food packaging materials.

The numbering was added to the Section signaled by the Reviewer and to other Sections where it was missing. Additionally, as asked, the content of this subsection has been enlarged by discussing the potential use of lignin nanoparticles to evolve sustainable food packaging materials. Please, see lines 743-762.

7) “5. Conclusions” (lines 822-864). This section perfectly remarks the most relevant outcomes found by the authors in this field and the promising future prospectives. It may be desirable to add a brief statement to also highlight the future action lines to pursue the topic covered in this topic.

As asked, a brief statement to highlight the future action lines to pursue nanomaterials has been included in the Conclusions. Please, see lines 1040-1056.

Round 2

Reviewer 2 Report

Comments and Suggestions for Authors

I suggested to re-structure the paper to be more standard and scientifically relevant.

You added 8 rows.

What I suggested was to follow this:

https://static1.squarespace.com/static/65b880e13b6ca75573dfe217/t/65d818f02bbbc04c85371122/1708660977279/PRISMA_2020_expanded_checklist.pdf

However, your paper was selected as an "article" rather than a "review." It is a hybrid product. Please, answer clearly and honestly to my review. Did I misunderstand your paper? Why did you structure it as this? Or was my suggestion not clear?

A reviewing process is a dialog. I am not "angry" because you didn't change the structure, but I am disappointed that you didn't answer clearly and explain your point of view. We are all researchers, so you don't need to 'check a box' without the content.

Author Response

I suggested to re-structure the paper to be more standard and scientifically relevant.

You added 8 rows.

What I suggested was to follow this:

https://static1.squarespace.com/static/65b880e13b6ca75573dfe217/t/65d818f02bbbc04c85371122/1708660977279/PRISMA_2020_expanded_checklist.pdf

However, your paper was selected as an "article" rather than a "review." It is a hybrid product. Please, answer clearly and honestly to my review. Did I misunderstand your paper? Why did you structure it as this? Or was my suggestion not clear?

A reviewing process is a dialog. I am not "angry" because you didn't change the structure, but I am disappointed that you didn't answer clearly and explain your point of view. We are all researchers, so you don't need to 'check a box' without the content

Dear Reviewer, sorry for not giving you further explanations (probably due), as to why we did not apply the PRISMA guidelines that you suggested, limiting to adding Section 1.3. I mistakenly thought they were not necessary. Indeed, your explicit request to insert a “Materials and methods" to present how we chose papers to include in the review, inclusion and exclusion criteria, what we analysed, etc. seemed to us the most essential, while the indication on PRISMA only a suggestion. Anyway, the Reviewer is right. We should have provided more explanations for our choices, already in the first-round revision, but we are still willing to provide them now. Concerning the suggestion of using PRISMA 2020, we did not consider it suitable for our review, which is not a “scoping review” or a “systematic review”, but rather a “narrative review” or more generally a “review” belonging to the “traditional review family”, as indicated before the title (https://doi.org/10.1111/hir.12276). In this regard, it has been organized according to the Instruction for Authors of Nanomaterials and other MDPI journals. PRISMA 2020 provides indications for the realization of reviews not belonging to the “traditional review family”, such as “systematic review” or “scoping reviews”, and Nanomaterials indicates the use of an extension of PRISMA only for “scoping review” (From Instruction for Authors of Nanomaterials: “A Scoping Review type can be submitted as a Review. The structure is similar to that of a review. Scoping reviews should strictly follow the PRISMA extension for scoping reviews checklist (https://www.prisma-statement.org/scoping) and submit the checklist as non-published material during submission. Templates for the flow diagram can be downloaded from the PRISMA website and the diagram should be included in the main text. We strongly encourage authors to register their detailed protocols, before data extraction commences, in a public registry such as the Open Science Framework (https://osf.io/) or Inplasy (https://inplasy.com/). Authors must include a statement about following the PRISMA guidelines and registration information (if available) in the Methods section.as a useful tool for the good realization of Reviews to be published”). Ours is not a “scoping review”, so it does not necessarily have to follow the structural indications of PRISMA 2020. Over the last 7 years, we have published several reviews in different relevant MDPI journals (Polymers, Nanomaterials, IJMS, Fibers, Antioxidants) and not only, following structural criteria like those used here, with much success and many citations received. Therefore, to answer the Reviewer's questions, I think that the Reviewer have misunderstood our paper, which has been structured as this, based on Nanomaterials instructions for “reviews” and the success obtained with previous reviews submitted to MDPI journals, and not only, including Nanomaterials, both among the Reviewers and among readers, once published. The Reviewer request appeared to me clear, but perhaps it was not, and it was not fully addressed in the previous round-revision. We apologise for this our misunderstanding. We hope to have addressed now all Reviewer concerns and requests. To give further reassurance to the Reviewer regarding the quality of this manuscript and of its structure, we make kindly note him/her, that the only other Reviewer out of 4, who seemed to have further comments in this second-round revision, as comment, he/she asserted to strongly endorse our manuscript for publication.

Reviewer 4 Report

Comments and Suggestions for Authors

The authors did a great deal of effort to cover all the suggestions raised by the Reviewers. For this reason, the scientific manuscript quality was greatly improved. Based on the significance of this Review work in the field, I warmly endorse it for further publication in Nanomaterials.

Author Response

The authors did a great deal of effort to cover all the suggestions raised by the Reviewers. For this reason, the scientific manuscript quality was greatly improved. Based on the significance of this Review work in the field, I warmly endorse it for further publication in Nanomaterials

We thank a lot the Reviewer for his/her great contribute to enhancing the scientific quality of this

work, for having appreciated our work of revision, and for her/his positive final comments.